# Evaluating the Generalization Ability of Quantized LLMs: Benchmark, Analysis, and Toolbox

## Abstract

Large language models (LLMs) have exhibited exciting progress in multiple scenarios, while the huge computational demands hinder their deployments in lots of real-world applications. As an effective means to reduce memory footprint and inference cost, quantization also faces challenges in performance degradation at low bit-widths. Understanding the impact of quantization on LLM capabilities, especially the *generalization ability*, is crucial. However, the community's main focus remains on the algorithms and models of quantization, with insufficient attention given to to the impact of *data* on the generalization abilities of quantized LLMs. In this work, we fill this gap by providing a comprehensive benchmark suite for this research topic, including an evaluation system, detailed analyses, and a general toolbox. Specifically, based on the dominant pipeline in LLM quantization, we primarily explore the impact of calibration data distribution on the generalization of quantized LLMs and conduct the benchmark using more than 40 datasets within two main scenarios. Based on this benchmark, we conduct extensive experiments with well-known LLMs (LLaMA and Baichuan) and four quantization algorithms to investigate this topic in-depth, yielding several counter-intuitive and valuable findings, *e.g.*, models quantized using a calibration set with the same distribution as the test data are not necessarily optimal. Besides, to facilitate future research, we also release a modular-designed toolbox, which decouples the overall pipeline into several separate components, *e.g.*, base LLM module, dataset module, quantizer module, *etc.* and allows subsequent researchers to easily assemble their methods through a simple configuration. Our code is submitted in the supplementary materials and will be publicly available.

## 1 Introduction

In recent years, large language models (LLMs) have made groundbreaking advancements, demonstrating remarkable results and outstanding *generalization ability* across various tasks (Vaswani, 2017; Zhang et al., 2022; Achiam et al., 2023; Touvron et al., 2023). For example, given a few prompt examples or questions, LLMs can produce insightful answers within the unseen domain (Radford et al., 2019; Brown et al., 2020). However, while LLMs exhibit remarkable capabilities, their substantial size makes real-world implementation cost-prohibitive. To address this challenge, model quantization has emerged as a prevailing technique for reducing the memory footprint of LLMs (Frantar et al., 2022; Lin et al., 2023; Dettmers et al., 2023; Chee et al., 2024; Yao et al., 2022; Xiao et al., 2023; Shao et al., 2023). Specifically, quantization reduces the model size by replacing high-precision floating-point numbers with lower-precision integers (*e.g.*, from FP16 to INT4) (Nagel et al., 2021; Gholami et al., 2022; Zhu et al., 2023). Currently, to avoid the substantial retraining costs of LLMs, the quantization methods for large models primarily employ post-training quantization (PTQ) (Frantar et al., 2022; Lin et al., 2023; Dettmers et al., 2023; Chee et al., 2024), which leverages calibration data to optimize the error caused by the quantization. Given the prevalent view that LLM capabilities stem from their extensive parameter count (Kaplan et al., 2020), a critical question emerges:

*Can the quantized LLMs still retain their strong generalization ability?*

While some works have acknowledged this issue (Liu et al., 2023a; Jaiswal et al., 2023; Li et al., 2024; Huang et al., 2024a; Jin et al., 2024b), there is still a lack of systematic evaluation regarding

the generalization performance of LLMs after quantization, particularly considering the impact of *calibration data* introduced during the quantization process.

As shown in Fig. 1, the process of model quantization encompasses three distinct stages: pre-training, quantization, and inference, utilizing pre-training data, calibration data, and test data, respectively. Existing quantization researches typically use a standard calibration set, which is usually a subset of the pre-training data (Scenario 1, **S1**), and evaluate on several fixed datasets (Paperno et al., 2016; Clark et al., 2018; Tata & Patel, 2003; Sakaguchi et al., 2021; Zellers et al., 2019). However, because using task-specific data for model calibration is a more reasonable choice in practical applications, the relationship between the distribution of *calibration data* and *test data* and its impact on the generalization ability of quantized models is a more worthy research topic that has not been deeply explored (Scenario 2, **S2**). In this work, to answer the abovementioned question and bridge the gap between academic research and practical implementation, we provide a platform to evaluate the generalization ability of quantized LLMs, covering *benchmarks*, *analyses*, and a modular-designed *toolbox*.

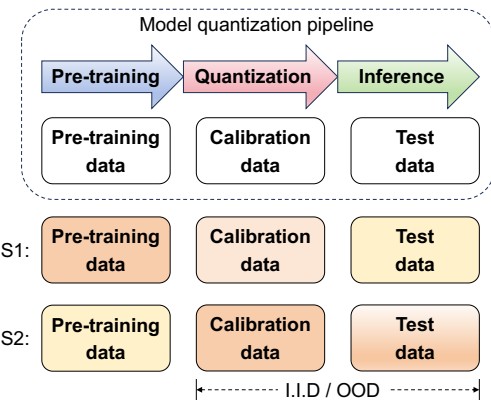

Figure 1: We show the pipeline of model quantization and the data required at each stage *(Top)*. The calibration data used in previous works generally share the same distribution with pre-training data *(S1)*, and the relation between calibration data and test data should be further discussed *(S2)*.

**Benchmark evaluation.** As shown in Fig. 1, we build the benchmark based on the two scenarios:

• In **S1** (Section 2), beyond the existing research, we collect the most comprehensive evaluation of test datasets to date, covering 9 categories and 26 datasets. We use C4 (Raffel et al., 2020) as the calibration dataset and quantize LLaMA2-7B (Touvron et al., 2023) model by two methods (Frantar et al., 2022; Dettmers et al., 2023) across three weight bit-widths.

• In **S2** (Section 3), our benchmark covers 19 datasets with two types of distribution shifts between calibration data and test data: *cross-dataset* and *cross-subject*. We consider both English and Chinese domains for the cross-dataset setting. Besides, our benchmark also includes a more challenging cross-subject setting, *e.g.* from humanities to social science. To our knowledge, no prior work has investigated the generalization of quantized models in a cross-subject setting. For all settings, our benchmark builds the Independent and Identically Distribution (I.I.D) and Out-of-Distribution (OOD) evaluations by adjusting the calibration data distributions. In our experiments, we quantize LLaMA family (Touvron et al., 2023) and Baichuan2-7B-Base (Du et al., 2021) for English and Chinese models with four methods (Frantar et al., 2022; Dettmers et al., 2023; Lin et al., 2023; Xiao et al., 2023) across three weight bit-widths.

The generalization performance of quantized models is assessed using zero-shot and few-shot evaluation for all experiments, and we summarize the key features of our benchmark in Tab. 1.

Based on experimental results, we provide the following answers to the core question of the paper.

**Answers.** Quantized LLMs maintain strong generalization capabilities under high bit-width scenarios. However, their generalization performance deteriorates significantly in extremely low bit-width scenarios. Additionally, we find that using calibration data related to the test data does not significantly enhance generalization performance.

**Empirical findings.** Based on the experiments, we observe several counter-intuitive phenomena, *e.g.*,

• *Tasks vary significantly in their sensitivity to quantization; low-bit quantization can lead to improved performance for certain tasks.* Scientific knowledge QA, reading comprehension, common sense reasoning, and mathematical reasoning are highly sensitive to quantization, particularly in low-bit scenarios, where quantization can lead to a significant decline in performance. Meanwhile, sentiment analysis is also very sensitive to quantization, but in low-bit situations, quantization can actually

Table 1: Summary of the proposed benchmark.

| Scenario | Distribution Shift | Task Language | Weight Precision | Model | Benchmark & Dataset | Results |
|---|---|---|---|---|---|---|
| S1 | – | English | {16, 4, 3, 2} | LLaMA2-7B | WinoGrande, WSC273, HellaSwag, SWAG, PIQA, MathQA, Mutual, Mutual_Plus, CrowS-Pairs, Toxigen, PubMedQA OpenBookQA, SciQ,ARC-Easy, ARC-Challenge, MC-TACO, RACE,QA4MRE, GLUE (6 datasets), ANLI, BLiMP | Fig. 2 |
| S2 | Cross-dataset | English | {4, 3} | LLaMA family | BOSS (16 datasets) | Tab. 2 |
| S2 | Cross-dataset | Chinese | {4, 3, 2} | Baichuan2-7B | C-EVAL, CMMLU | Tab. 13 |
| S2 | Cross-discipline | Chinese | {4, 3, 2} | Baichuan2-7B | C-EVAL | Tab. 14 |

lead to an improvement in performance. Tasks such as natural language inference demonstrate considerable robustness to quantization, with minimal changes in performance after quantization.

• *Consistency between calibration data and test distribution does not always yield optimal performance*, which is in stark contrast to the consensus during the pre-training and fine-tuning phases. The performance of quantized models using calibration data with distributions similar to or identical to those of downstream tasks is comparable to that achieved using subsets from high-quality corpora.

**Toolbox.** To support this work and facilitate future research, we develop a modular-designed code library. Specifically, this toolbox decouples the overall pipeline shown in Fig.1 into several separate components, *e.g.*, LLM module, dataset module, quantizer module, *etc.*, and provides common choices for each component and easy-to-use interface for possible extensions (see Section 4 and Fig. 4 for more details of the toolbox). This toolbox will be open-sourced along with the benchmark to facilitate future quantization applications and research.

## 2 S1: GENERALIZATION ASSESSMENT OF QUANTIZED LLMS WITH STANDARD SETTING

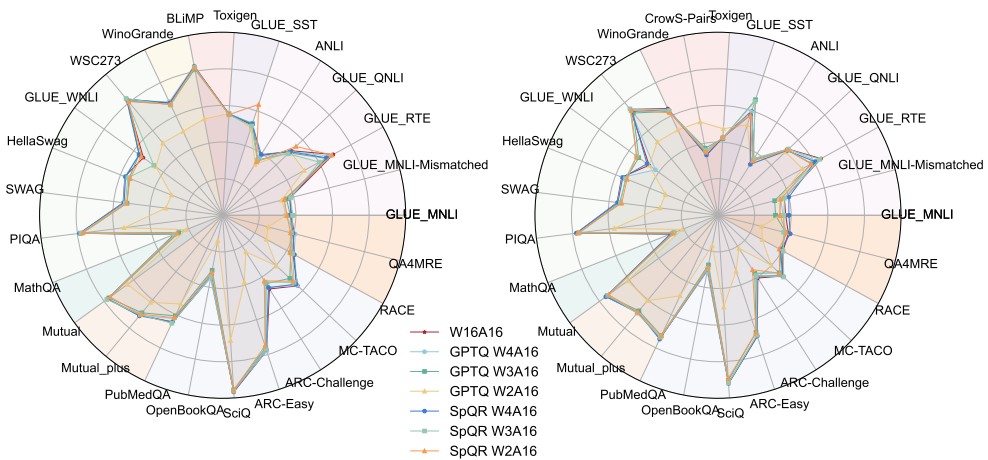

Figure 2: **S1**: Evaluation of quantized LLaMA2-7B on several standard datasets. Quantization methods include GPTQ and SpQR. Quantization bits include W4A16, W3A16, and W2A16, with W16A16 used as reference. The left figure shows 5-shot results, and the right figure shows 0-shot results. Different background colors represent different task types.

**Experiment settings**. To assess the difference in generalization ability, it is necessary to ensure that all other settings remain consistent except for the quantization process. To maintain consistency in the data encountered by the model before and after quantization, we strive to use calibration data during quantization that is as similar as possible to the data used during the pre-training phase of the LLM, namely the dataset C4 (Raffel et al., 2020) derived from pre-training data. The experimental setting is consistent with the evaluation settings used previously for quantized models (Frantar et al., 2022; Dettmers et al., 2023; Jaiswal et al., 2023; Liu et al., 2023a; Li et al., 2024). We utilize the LM Evaluation Harness (Gao et al., 2021a) with recommended parameters to conduct zero-shot and

few-shot tests on the following tasks. We provide full configurations in Appendix C, as well as code in the *supplementary materials*.

The 26 datasets we evaluated can be roughly divided into 9 categories: ❶common sense reasoning, ❷mathematical reasoning, ❸multi-turn dialogue reasoning, ❹bias diagnosis and mitigation, ❺scientific knowledge question answering, ❻reading comprehension, ❼natural language inference, ❽sentiment analysis, and ❾syntax phenomena evaluation. The common sense reasoning datasets include WinoGrande (Sakaguchi et al., 2021), WSC273 (Levesque et al., 2012), GLUE-WNLI (Wang et al., 2018), HellaSwag (Zellers et al., 2019), SWAG (Zellers et al., 2018), and PIQA (Tata & Patel, 2003). The mathematical reasoning datasets include MathQA(Amini et al., 2019). The multi-turn dialogue reasoning datasets include Mutual and Mutual_Plus (Cui et al., 2020). The bias diagnosis and mitigation datasets include CrowS-Pairs (Nangia et al., 2020) and Toxigen (Hartvigsen et al., 2022). The scientific knowledge question answering datasets include PubMedQA (Jin et al., 2019), OpenBookQA (Mihaylov et al., 2018), SciQ (Welbl et al., 2017), ARC-Easy, ARC-Challenge (Clark et al., 2018), and MC-TACO (Zhou et al., 2019). The reading comprehension datasets include RACE (Lai et al., 2017) and QA4MRE (Peñas et al., 2013). The natural language inference datasets include GLUE-MNLI, GLUE-MNLI-Mismatched, GLUE-RTE, GLUE-QNLI (Wang et al., 2018), and ANLI (Nie et al., 2019). The sentiment analysis dataset includes GLUE-SST (Wang et al., 2018). The syntax phenomena evaluation dataset includes BLiMP (Warstadt et al., 2020).

**Results and analysis**. We present the experimental results for both the 5-shot and 0-shot scenarios in Fig. 2. To more clearly observe the experimental results of different downstream task types, we average the decline in accuracy after quantization per task type. We present the results in Tab. 12 and conduct a more detailed analysis in Appendix D.2. It can be obviously observed that *models still retain strong generalization capabilities under low-bit quantization, but experience significant performance degradation under ultra-low-bit quantization*. When quantizing model weights to 3-4 bits, the performance degradation of all methods is not very pronounced. For some datasets, quantizing to 4 bits even leads to higher model performance compared to full precision. However, when weights are quantized to 2 bits, GPTQ exhibits a significant performance drop on most tasks. Compared to other methods, SPQR maintains relatively good performance at 2 bits, which may be attributed to SPQR's ability to identify and isolate outlier weights. Additionally, *different tasks exhibit varying sensitivities to quantization*, with scientific knowledge QA, reading comprehension, common sense reasoning, mathematical reasoning and sentiment analysis showing higher sensitivity while tasks like natural language inference emerge lower sensitivity. For example, in extreme quantization scenarios in Tab. 12, the performance of scientific knowledge QA declines by up to 40%, while the performance drop for natural language inference is only 10%. Interestingly, we observe sentiment analysis actually shows a significant performance improvement under extreme quantization scenarios. We also demonstrate the robustness of the experiments with respect to random seeds in Appendix D.1.

# 3 S2: GENERALIZATION ASSESSMENT OF QUANTIZED LLMS WITH DOMAIN SHIFTS

This section investigates novel generalization scenarios in quantization, where different generalization scenarios serve as instantiations of the framework. The distribution shift we consider primarily pertains to the shift from calibration data to test data. Types of distribution shift include *cross-dataset* distribution shift and *cross-subject* distribution shift, aimed at studying the impact of distribution shift from calibration data to test data on quantized model performance. Cross-dataset distribution shift refers to using different datasets as calibration set, while cross-subject distribution shift refers to using different subjects from the same dataset as calibration set. Experiments will encompass two main categories: *English cross-dataset* distribution shift experiments on the out-of-distribution generalization benchmark BOSS in Sec. 3.1, and *Chinese cross-dataset* distribution shift experiments as well as *Chinese cross-subject* distribution shift experiments on Chinese domain-specific tasks in Sec. 3.2. We will conduct an in-depth analysis of the results from S2 in Sec. 3.3. We provide full configurations in Appendix C, as well as code in the *supplementary materials*.

## 3.1 ENGLISH CROSS-DATASET TRANSFER TASK

**Experiment settings**. We evaluate *cross-dataset* distribution shift experiments on the OOD benchmark BOSS (Yuan et al., 2024) in NLP. Previous work in NLP concerning OOD mostly considers

Table 2: Cross-dataset distribution shift evaluation on BOSS. "Calib." represents the calibration dataset, and "Gene." represents generalization scenario. To save space, abbreviations are used for datasets. Each row presents experimental results using different datasets as calibration sets on the same test dataset. Results with blue backgrounds indicate I.I.D results, while those without color represent OOD results. The higher the metric, the better the performance. Bold results indicate the best performance on the same test dataset. The result indicates I.I.D dataset achieve the best performance. Note: "-" indicates out of memory results.

| Method | Test | Gene. | W/A | SQ | AQA | NQA | SQA | Test | Gene. | W/A | AZ | DS | SE | SST | Test | Gene. | W/A | MN | AN | WN | CN | Test | Gene. | W/A | CC | AC | IH | TG |
|---|---|---|---|---|---|---|---|---|---|---|---|---|---|---|---|---|---|---|---|---|---|---|---|---|---|---|---|---|---|
| GPTQ | SQ | 0-shot | 4/16 | 53.84 | 52.73 | 54.69 | **57.31** | AZ | 0-shot | 4/16 | 70.81 | 17.87 | 63.18 | **72.08** | MN | 0-shot | 4/16 | **0.36** | 0.23 | 0.22 | - | CC | 0-shot | 4/16 | 23.90 | 26.96 | 52.52 | **53.32** |
| | | | 3/16 | 45.31 | 48.86 | 49.49 | **50.79** | | | 3/16 | **38.06** | 0.38 | 0.26 | 0.04 | | | 3/16 | 0.00 | 0.00 | 0.00 | - | | | 3/16 | 0.60 | 2.45 | 9.70 | **10.60** |
| | | 1-shot | 4/16 | 67.04 | 65.97 | 67.06 | **68.16** | | 3-shot | 4/16 | **83.69** | 56.66 | 80.79 | 82.55 | | 3-shot | 4/16 | **49.69** | 32.81 | 34.93 | - | | 2-shot | 4/16 | 91.80 | 87.46 | 91.71 | **91.84** |
| | | | 3/16 | 60.76 | 58.84 | 63.34 | 63.01 | | | 3/16 | **74.54** | 24.86 | 59.06 | 59.79 | | | 3/16 | **34.12** | 31.79 | 31.82 | - | | | 3/16 | 89.11 | 35.94 | **91.96** | 90.35 |
| | AQA | 0-shot | 4/16 | 28.00 | 27.12 | 28.40 | **30.40** | DS | 0-shot | 4/16 | 46.10 | 21.37 | 31.82 | **46.79** | AN | 0-shot | 4/16 | **1.07** | 0.52 | 0.93 | - | AC | 0-shot | 4/16 | **19.12** | 5.93 | 7.84 | 17.02 |
| | | | 3/16 | 21.81 | **25.28** | 23.35 | 24.99 | | | 3/16 | **17.59** | 1.72 | 0.01 | 0.00 | | | 3/16 | **4.17** | 0.00 | 0.00 | - | | | 3/16 | 0.76 | **1.72** | 0.19 | 0.57 |
| | | 1-shot | 4/16 | 35.50 | **36.11** | 31.97 | 35.77 | | 3-shot | 4/16 | 54.40 | 38.78 | 52.54 | **55.50** | | 3-shot | 4/16 | **34.34** | 33.76 | 33.24 | - | | 2-shot | 4/16 | 15.87 | **17.59** | 15.87 | 16.25 |
| | | | 3/16 | 31.39 | 29.54 | 31.60 | **32.24** | | | 3/16 | 54.68 | 36.05 | 33.86 | 43.46 | | | 3/16 | 30.97 | **33.69** | 33.28 | - | | | 3/16 | 60.23 | **90.35** | 15.87 | 56.02 |
| | NQA | 0-shot | 4/16 | 37.94 | **38.76** | 38.63 | 38.23 | SE | 0-shot | 4/16 | 18.32 | 8.21 | 15.60 | **26.43** | WN | 0-shot | 4/16 | 0.09 | 0.04 | **0.11** | - | IH | 0-shot | 4/16 | 37.37 | 22.55 | 33.90 | **40.82** |
| | | | 3/16 | 31.36 | 33.79 | 33.37 | **34.45** | | | 3/16 | **4.83** | 0.09 | 0.20 | 0.01 | | | 3/16 | **0.49** | 0.00 | 0.00 | - | | | 3/16 | 11.27 | 7.32 | 4.53 | **13.18** |
| | | 1-shot | 4/16 | 48.55 | 49.30 | **49.73** | 49.09 | | 3-shot | 4/16 | 42.96 | 28.55 | 42.99 | **44.75** | | 3-shot | 4/16 | 41.51 | 43.34 | **47.53** | - | | 2-shot | 4/16 | 62.36 | **63.46** | 62.00 | 62.29 |
| | | | 3/16 | 44.38 | 43.35 | **46.95** | 45.61 | | | 3/16 | 42.36 | 22.67 | 35.54 | 29.40 | | | 3/16 | 38.83 | 48.09 | **48.15** | - | | | 3/16 | 63.52 | **90.35** | 61.83 | 61.77 |
| | SQA | 0-shot | 4/16 | 42.58 | 45.72 | **46.21** | 44.20 | SST | 0-shot | 4/16 | **49.15** | 20.73 | 27.12 | 44.98 | CN | 0-shot | 4/16 | **0.06** | 0.00 | 0.00 | - | TG | 0-shot | 4/16 | 48.44 | 36.72 | 44.84 | **57.97** |
| | | | 3/16 | 30.19 | 26.99 | 28.49 | **33.73** | | | 3/16 | 7.82 | 1.04 | 0.00 | 0.00 | | | 3/16 | 0.06 | 1.12 | **1.45** | - | | | 3/16 | 12.81 | 9.53 | 2.19 | **14.06** |
| | | 1-shot | 4/16 | 56.04 | 61.89 | 60.92 | **62.17** | | 3-shot | 4/16 | **60.50** | 33.25 | 45.24 | 51.11 | | 3-shot | 4/16 | 35.23 | **36.35** | 32.44 | - | | 2-shot | 4/16 | 72.03 | **75.47** | 67.81 | 68.40 |
| | | | 3/16 | 43.46 | 42.83 | 45.17 | **48.82** | | | 3/16 | **54.37** | 33.25 | 35.46 | 50.20 | | | 3/16 | 29.54 | 29.03 | **33.39** | - | | | 3/16 | 70.47 | **90.35** | 57.50 | 62.19 |
| SpQR | SQ | 0-shot | 4/16 | **57.03** | 49.87 | 53.00 | 54.36 | AZ | 0-shot | 4/16 | 63.34 | 62.46 | 72.52 | **83.14** | MN | 0-shot | 4/16 | **0.57** | 0.02 | 0.13 | - | CC | 0-shot | 4/16 | **61.73** | 59.48 | 58.92 | 37.48 |
| | | | 3/16 | 52.37 | 45.90 | 54.55 | **58.36** | | | 3/16 | **72.38** | 55.79 | 37.28 | 27.84 | | | 3/16 | 0.00 | **0.01** | 0.00 | - | | | 3/16 | **36.90** | 2.54 | 15.42 | 22.38 |
| | | 1-shot | 4/16 | 66.45 | 66.80 | **67.41** | 67.21 | | 3-shot | 4/16 | 79.65 | 69.31 | **85.44** | 82.91 | | 3-shot | 4/16 | 36.19 | 40.45 | **41.62** | - | | 2-shot | 4/16 | **90.65** | 89.27 | 91.74 | 84.69 |
| | | | 3/16 | 65.12 | 65.55 | **68.65** | 66.95 | | | 3/16 | 83.68 | **86.30** | 72.18 | 83.50 | | | 3/16 | 32.39 | **40.31** | 38.47 | - | | | 3/16 | 87.70 | **91.76** | 86.99 | 83.56 |
| | AQA | 0-shot | 4/16 | **30.59** | 25.11 | 27.60 | 29.50 | DS | 0-shot | 4/16 | 35.47 | 43.53 | 40.85 | **50.40** | AN | 0-shot | 4/16 | **0.86** | 0.07 | 0.28 | - | AC | 0-shot | 4/16 | 10.13 | 4.97 | 12.05 | **13.58** |
| | | | 3/16 | 26.35 | 21.43 | 27.55 | **30.36** | | | 3/16 | **41.87** | 31.17 | 15.42 | 29.10 | | | 3/16 | 0.00 | **0.07** | 0.00 | - | | | 3/16 | 2.49 | 0.76 | **7.84** | 2.87 |
| | | 1-shot | 4/16 | **37.64** | 36.63 | 36.94 | 35.42 | | 3-shot | 4/16 | 50.82 | 46.67 | **57.74** | 56.34 | | 3-shot | 4/16 | 33.17 | 33.31 | **33.79** | - | | 2-shot | 4/16 | 16.44 | **21.03** | 15.87 | 20.46 |
| | | | 3/16 | 34.61 | 34.75 | **37.49** | 33.10 | | | 3/16 | **59.10** | 54.80 | 52.56 | 56.02 | | | 3/16 | **33.66** | 31.93 | 33.14 | - | | | 3/16 | 15.87 | 15.87 | **19.31** | 15.87 |
| | NQA | 0-shot | 4/16 | **40.30** | 38.01 | 39.40 | 38.22 | SE | 0-shot | 4/16 | 14.62 | 23.36 | 19.85 | **33.24** | WN | 0-shot | 4/16 | **0.28** | 0.00 | 0.00 | - | IH | 0-shot | 4/16 | **42.21** | 41.79 | 40.12 | 31.76 |
| | | | 3/16 | 35.79 | 33.27 | **40.80** | 38.77 | | | 3/16 | **16.05** | 10.22 | 4.75 | 7.30 | | | 3/16 | 0.00 | **0.06** | 0.00 | - | | | 3/16 | **31.32** | 6.78 | 17.68 | 16.96 |
| | | 1-shot | 4/16 | 49.61 | 49.12 | **49.70** | 48.47 | | 3-shot | 4/16 | **44.48** | 44.15 | 44.25 | 44.39 | | 3-shot | 4/16 | 43.28 | **43.77** | 41.79 | - | | 2-shot | 4/16 | 64.24 | **65.85** | 62.14 | 66.07 |
| | | | 3/16 | 48.25 | 46.61 | **48.99** | 47.79 | | | 3/16 | **53.16** | 43.63 | 41.76 | 44.77 | | | 3/16 | 39.09 | **47.32** | 40.77 | - | | | 3/16 | 62.95 | 63.14 | **63.17** | 64.37 |
| | SQA | 0-shot | 4/16 | **46.45** | 42.62 | 44.30 | 45.10 | SST | 0-shot | 4/16 | 46.02 | 29.47 | 44.72 | **55.67** | CN | 0-shot | 4/16 | 0.00 | 0.22 | **0.45** | - | TG | 0-shot | 4/16 | **54.37** | 52.66 | 51.09 | 39.53 |
| | | | 3/16 | 36.90 | **44.57** | 42.88 | 39.31 | | | 3/16 | **23.08** | 14.87 | 3.65 | 6.52 | | | 3/16 | 0.06 | 0.00 | **0.89** | - | | | 3/16 | **41.88** | 9.69 | 19.38 | 37.34 |
| | | 1-shot | 4/16 | 61.63 | 57.77 | **61.79** | 60.55 | | 3-shot | 4/16 | 55.41 | 42.37 | 58.54 | **59.32** | | 3-shot | 4/16 | **36.13** | 34.84 | 34.23 | - | | 2-shot | 4/16 | 69.84 | 76.56 | 61.41 | **77.60** |
| | | | 3/16 | 48.86 | **59.19** | 56.34 | 55.06 | | | 3/16 | **63.49** | 60.37 | 53.98 | 61.80 | | | 3/16 | 35.29 | **35.90** | 33.17 | - | | | 3/16 | 73.13 | 66.88 | 68.44 | **77.03** |
| AWQ | SQ | 0-shot | 4/16 | **56.73** | 55.09 | 52.09 | 50.21 | AZ | 0-shot | 4/16 | - | 5.42 | **35.23** | 33.65 | MN | 0-shot | 4/16 | **0.48** | 0.14 | 0.06 | - | CC | 0-shot | 4/16 | 50.17 | **66.60** | 42.19 | 42.11 |
| | | | 3/16 | **48.32** | 37.95 | 44.45 | 40.30 | | | 3/16 | - | 39.41 | **70.10** | 35.95 | | | 3/16 | 0.00 | 0.01 | 0.01 | - | | | 3/16 | 41.96 | 39.03 | **46.95** | 14.72 |
| | | 1-shot | 4/16 | 66.57 | 66.91 | **67.02** | 66.21 | | 3-shot | 4/16 | - | 83.64 | **83.73** | 78.06 | | 3-shot | 4/16 | **42.20** | 38.37 | 36.05 | - | | 2-shot | 4/16 | **91.84** | 91.63 | 90.80 | 89.31 |
| | | | 3/16 | 59.81 | **61.81** | 61.27 | 61.38 | | | 3/16 | - | 88.73 | **90.16** | 88.92 | | | 3/16 | **35.44** | 34.22 | 35.34 | - | | | 3/16 | 36.43 | 73.04 | **90.93** | 27.24 |
| | AQA | 0-shot | 4/16 | **29.73** | 29.20 | 28.34 | 27.57 | DS | 0-shot | 4/16 | - | 2.36 | 20.10 | **22.19** | AN | 0-shot | 4/16 | **0.59** | 0.07 | 0.07 | - | AC | 0-shot | 4/16 | 9.56 | **11.85** | 11.28 | 5.55 |
| | | | 3/16 | **23.02** | 17.58 | 20.37 | 18.62 | | | 3/16 | - | 8.76 | **27.09** | 11.87 | | | 3/16 | 0.00 | **0.07** | 0.00 | - | | | 3/16 | **5.74** | 4.59 | 4.21 | 1.15 |
| | | 1-shot | 4/16 | 35.76 | 37.01 | **37.55** | 36.78 | | 3-shot | 4/16 | - | 53.91 | **55.92** | 50.95 | | 3-shot | 4/16 | **33.66** | 33.66 | 33.66 | - | | 2-shot | 4/16 | 15.87 | 15.87 | 16.06 | **16.63** |
| | | | 3/16 | 31.64 | 33.04 | 32.88 | **33.46** | | | 3/16 | - | 50.95 | 56.24 | **59.05** | | | 3/16 | **33.69** | 32.55 | 33.69 | - | | | 3/16 | 24.86 | 18.93 | 16.06 | **56.02** |
| | NQA | 0-shot | 4/16 | 39.20 | 38.58 | **39.47** | 38.10 | SE | 0-shot | 4/16 | - | 4.19 | **18.90** | 14.96 | WN | 0-shot | 4/16 | **0.30** | 0.17 | 0.02 | - | IH | 0-shot | 4/16 | 37.59 | **44.64** | 34.09 | 27.16 |
| | | | 3/16 | **35.75** | 31.27 | 32.91 | 33.69 | | | 3/16 | - | 5.52 | **14.95** | 5.49 | | | 3/16 | 0.00 | 0.00 | 0.00 | - | | | 3/16 | 20.22 | 17.97 | **25.72** | 4.73 |
| | | 1-shot | 4/16 | 43.25 | 43.18 | **43.39** | 42.56 | | 3-shot | 4/16 | - | 45.03 | **45.44** | 43.77 | | 3-shot | 4/16 | **40.02** | 39.40 | 38.23 | - | | 2-shot | 4/16 | 62.36 | 62.46 | **65.03** | 64.67 |
| | | | 3/16 | 41.02 | 40.50 | **41.27** | 41.26 | | | 3/16 | - | 38.53 | **55.02** | 44.50 | | | 3/16 | 37.11 | **44.38** | 37.17 | - | | | 3/16 | 61.85 | **63.03** | 61.88 | 61.79 |
| | SQA | 0-shot | 4/16 | 43.83 | 43.07 | **44.32** | 44.20 | SST | 0-shot | 4/16 | - | 2.09 | 11.47 | **19.17** | CN | 0-shot | 4/16 | 3.35 | 1.56 | **3.41** | - | TG | 0-shot | 4/16 | 49.38 | **52.5** | 40.31 | 36.56 |
| | | | 3/16 | **35.10** | 29.62 | 29.55 | 32.07 | | | 3/16 | - | 3.39 | **30.77** | 8.21 | | | 3/16 | **3.07** | 0.06 | 1.79 | - | | | 3/16 | 26.72 | 20.00 | **37.03** | 8.44 |
| | | 1-shot | 4/16 | 48.12 | 48.39 | **49.37** | 47.24 | | 3-shot | 4/16 | - | 58.28 | **58.80** | 51.76 | | 3-shot | 4/16 | 33.84 | **34.51** | 33.28 | - | | 2-shot | 4/16 | 65.31 | 65.47 | 71.25 | **75.16** |
| | | | 3/16 | 40.48 | 39.14 | 39.84 | **43.61** | | | 3/16 | - | 57.11 | 64.93 | **65.84** | | | 3/16 | 28.14 | **33.61** | 29.87 | - | | | 3/16 | 68.75 | **74.22** | 63.91 | 67.66 |
| SQ | SQ | 0-shot | 4/8 | 35.34 | 39.17 | 40.12 | **40.64** | AZ | 0-shot | 4/8 | 0.00 | **0.87** | 0.00 | 0.01 | MN | 0-shot | 4/8 | **0.01** | 0.00 | 0.00 | - | CC | 0-shot | 4/8 | **0.03** | 0.00 | 0.00 | 0.00 |
| | | | 3/8 | **0.01** | 0.01 | 0.00 | 0.01 | | | 3/8 | 0.00 | 0.00 | 0.00 | 0.00 | | | 3/8 | 0.00 | 0.00 | 0.00 | - | | | 3/8 | 0.00 | 0.00 | **0.01** | 0.00 |
| | | 1-shot | 4/8 | 28.01 | 56.13 | 55.12 | **56.59** | | 3-shot | 4/8 | 54.76 | **88.36** | 85.90 | 84.85 | | 3-shot | 4/8 | 31.70 | 32.86 | **34.54** | - | | 2-shot | 4/8 | **90.39** | 65.29 | 65.08 | 87.63 |
| | | | 3/8 | 0.00 | 0.00 | 0.00 | 0.01 | | | 3/8 | 0.00 | 0.00 | 0.00 | 0.00 | | | 3/8 | 0.00 | 0.00 | 0.00 | - | | | 3/8 | 0.00 | 0.00 | 0.00 | 0.00 |
| | AQA | 0-shot | 4/8 | 14.22 | 18.10 | **18.18** | 18.17 | DS | 0-shot | 4/8 | 0.00 | **0.02** | 0.00 | 0.00 | AN | 0-shot | 4/8 | **0.03** | 0.00 | 0.00 | - | AC | 0-shot | 4/8 | **0.19** | 0.19 | 0.19 | 0.00 |
| | | | 3/8 | **0.01** | 0.00 | 0.00 | 0.01 | | | 3/8 | 0.00 | 0.00 | 0.00 | 0.00 | | | 3/8 | 0.00 | 0.00 | 0.00 | - | | | 3/8 | 0.00 | 0.00 | 0.00 | 0.00 |
| | | 1-shot | 4/8 | 28.01 | 28.91 | 27.96 | **29.13** | | 3-shot | 4/8 | 47.01 | **50.42** | 50.00 | 34.88 | | 3-shot | 4/8 | 32.97 | **33.93** | 33.14 | - | | 2-shot | 4/8 | 18.16 | 23.71 | **53.15** | 23.33 |
| | | | 3/8 | 0.00 | 0.00 | 0.00 | 0.00 | | | 3/8 | 0.00 | 0.00 | 0.00 | 0.00 | | | 3/8 | 0.00 | 0.00 | 0.00 | - | | | 3/8 | 0.00 | 0.00 | 0.00 | 0.00 |
| | NQA | 0-shot | 4/8 | 24.26 | **27.83** | 23.95 | 24.07 | SE | 0-shot | 4/8 | 0.00 | 0.00 | 0.00 | **0.01** | WN | 0-shot | 4/8 | 0.00 | 0.00 | 0.00 | - | IH | 0-shot | 4/8 | **0.09** | 0.01 | 0.00 | 0.04 |
| | | | 3/8 | 0.00 | 0.00 | 0.00 | **0.01** | | | 3/8 | 0.00 | 0.00 | 0.00 | 0.00 | | | 3/8 | 0.00 | 0.00 | 0.00 | - | | | 3/8 | 0.00 | 0.00 | **0.02** | 0.00 |
| | | 1-shot | 4/8 | 30.69 | 32.16 | 29.83 | **33.18** | | 3-shot | 4/8 | 47.03 | 34.43 | 43.43 | 34.27 | | 3-shot | 4/8 | **47.32** | 46.70 | 47.15 | - | | 2-shot | 4/8 | **61.87** | 60.73 | 59.28 | 57.42 |
| | | | 3/8 | 0.00 | 0.00 | 0.00 | 0.00 | | | 3/8 | 0.00 | 0.00 | 0.00 | 0.00 | | | 3/8 | 0.00 | 0.00 | 0.00 | - | | | 3/8 | 0.00 | 0.00 | 0.00 | 0.00 |
| | SQA | 0-shot | 4/8 | 19.92 | **20.30** | 19.07 | 18.07 | SST | 0-shot | 4/8 | 0.00 | 0.00 | 0.00 | 0.00 | CN | 0-shot | 4/8 | 0.00 | 0.00 | 0.00 | - | TG | 0-shot | 4/8 | **0.63** | 0.00 | 0.16 | 0.00 |
| | | | 3/8 | 0.00 | 0.00 | 0.00 | **0.01** | | | 3/8 | 0.00 | 0.00 | 0.00 | 0.00 | | | 3/8 | 0.00 | 0.00 | 0.00 | - | | | 3/8 | 0.00 | 0.00 | **0.31** | 0.00 |
| | | 1-shot | 4/8 | **25.64** | 17.73 | 21.70 | 21.10 | | 3-shot | 4/8 | 26.47 | 53.06 | **55.02** | 36.90 | | 3-shot | 4/8 | 26.02 | **27.19** | 14.91 | - | | 2-shot | 4/8 | 58.28 | **65.31** | 59.06 | 59.38 |
| | | | 3/8 | 0.00 | 0.00 | 0.00 | 0.00 | | | 3/8 | 0.00 | 0.00 | 0.00 | 0.00 | | | 3/8 | 0.00 | 0.00 | 0.00 | - | | | 3/8 | 0.00 | 0.00 | 0.00 | 0.00 |

distribution shifts from various sources, *e.g.* from movies to Twitter (Yu et al., 2024). GLUE-X (Yang et al., 2022) and BOSS (Yuan et al., 2024) represent pioneering efforts in benchmarking OOD generalization in NLP. BOSS, building upon GLUE-X, improves by employing SimCSE scores for detection analysis and identifying dataset pairs exhibiting the lowest semantic similarity. These pairs are then utilized for training and testing, constructing a benchmark consisting of five downstream tasks. Each downstream task comprises an d I.I.D dataset and three OOD datasets.

To evaluate the generalization ability of quantized models in cross-dataset distribution shift experiments, we randomly sample 300 samples from the test set of each OOD dataset within the BOSS

benchmark as its corresponding training set, serving as the calibration set for the quantization process. For each downstream task, we utilize the training set from different datasets as the calibration set for the quantization process and test on the corresponding I.I.D and OOD test sets. In our experiments, we employ LLaMA2-7B, LLaMA2-13B and LLaMA3-8B (Touvron et al., 2023) as the target for quantization and selected four PTQ methods: GPTQ (Frantar et al., 2022), AWQ (Lin et al., 2023), SpQR (Dettmers et al., 2023), and SmoothQuant (Xiao et al., 2023). Given that there is not much difference in performance between excessively high bits and full precision, and too low a bit has already lost basic performance in these tasks, we only present the results of LLaMA2-7B weights quantizing to 3-4 bits with SmoothQuant quantizing the activations to 8 bits. Results for more models are in Appendix D.4, full precision results are in Appendix D.5 and 2-bit results are in Appendix D.6. We test two forms: 0-shot and few-shot.

**Results and Analysis**. We present the results in Tab. 2. We evaluate four downstream tasks in BOSS: EQA, SA, NLI, and TD. Each downstream task consists of four datasets, with each dataset tested using four datasets as calibration set. The following conclusions can be observed: *For the same test dataset, it's not necessarily the case that using I.I.D dataset as calibration set yield superior performance.* We can clearly observe that the overlap frequency between the I.I.D results with background and the bolded best performance results in Tab. 2 is not high, and may even be quite low. This suggests that the I.I.D dataset, when used as a calibration set, does not reliably enhance the capabilities of the quantized model. This is a counter-intuitive conclusion, as it is a well-established fact in fields such as pre-training and instruction fine-tuning that I.I.D datasets used as training sets can improve performance on corresponding downstream tasks. For instance, when testing SQ dataset in EQA task with GPTQ, the performance when using the SQ dataset as the calibration set is not ideal; on the contrary, using the SQA dataset as the calibration set yields better performance. We also demonstrate the robustness of the experiments with respect to random seeds in Appendix D.1.

## 3.2 Chinese Cross-dataset and Cross-Discipline Tasks

**Experiment settings**. We evaluate *cross-dataset* distribution shift experiments and *cross-subject* distribution shift experiments on the Chinese domain-specific datasets C-EVAL (Huang et al., 2024b) and CMMLU (Li et al., 2023). C-EVAL serves as a comprehensive benchmark for evaluating Chinese LLM. It consists of 13,948 multiple-choice questions covering 52 different subjects categorized into Humanities, Social Sciences, STEM, and Other. CMMLU is another Chinese evaluation dataset designed specifically to assess the advanced knowledge and reasoning abilities of LLM in the context of the Chinese language and culture. It encompasses 67 different subjects categorized into Humanities, Social Sciences, STEM, and Chinese specific and others.

Both C-EVAL and CMMLU, two Chinese-specific domain datasets, include Humanities, Social Sciences, and STEM three subject categories. We design cross-dataset distribution shift experiments based on the same subject categories. For each subject test, we respectively utilize the corresponding subjects from C-EVAL and CMMLU as calibration set to assess the impact of different datasets as calibration set on the test results. Additionally, we conducted cross-subject distribution shift experiments on the C-EVAL dataset. For each subject test, we use Humanities, Social Sciences, and STEM as calibration set to evaluate the influence of different subject subsets as calibration set on the test results. Since both C-EVAL and CMMLU lack training datasets, we used the validation dataset of C-EVAL as the training dataset and randomly sampled 300 samples from the test dataset of CMMLU as the training dataset. We utilize the Chinese LLM Baichuan2-7B-Base (Yang et al., 2023) as the quantization target and selecte four PTQ methods: GPTQ (Frantar et al., 2022), AWQ (Lin et al., 2023), SpQR (Dettmers et al., 2023), and SmoothQuant (Xiao et al., 2023). We quantize the weights to 2-4 bits, with SmoothQuant quantizing the activations to 8 bits, and test both 0-shot and 5-shot forms.

**Results and Analysis**. The results of cross-dataset distribution shift experiments on C-EVAL and CMMLU are presented in Tab. 13. We observed that *the performance of the I.I.D dataset is slightly higher than that of the OOD dataset, but it still tends to be random.* In most cases, the performance of the I.I.D dataset as a calibration set is better at times, while at other times the OOD dataset performs better, or both perform equally well. For example, when testing C-EVAL with GPTQ, the I.I.D dataset C-EVAL significantly outperforms CMMLU as the calibration set; however, when testing CMMLU with GPTQ, the performance of the I.I.D and OOD datasets tends to be random. There are slight differences in experimental results across different algorithms. For instance, when testing the

C-EVAL dataset with GPTQ, the I.I.D dataset C-EVAL serves as a better calibration set, whereas when testing the C-EVAL dataset with AWQ, the OOD dataset CMMLU appears to yield better performance for the quantized model.

The results of cross-subject distribution shift experiments on C-EVAL are presented in Tab. 14. *The results tend to be more random, and no conclusion can be drawn that using an I.I.D dataset as calibration set results in higher test accuracy.* An intuitive idea is that the subjects of humanities and social sciences are closely related and differ significantly from STEM, suggesting that using similar datasets would outperform those that are more disparate. However, our experimental results indicate that when testing on humanities and social sciences, there are instances where using STEM as a calibration set can also lead to higher performance. Similarly, when testing on STEM, there are occasions when using humanities and social sciences as calibration sets yields better performance than STEM itself. Thus, the conclusion that cross-disciplinary approaches result in a significant performance drop does not hold true.

### 3.3 S2: COMPARATIVE ANALYSIS OF ALL RESULTS

**Overall Findings.** *Consistency between calibration data and test distribution does not always yield optimal performance.* We integrate Tab. 2,Tab. 13and Tab. 14 and calculate the wining rates for two strategies: using I.I.D dataset and OOD dataset as the test set. To compare the model's performance when using I.I.D and OOD datasets as calibration sets, the calculation method for the IID wining rate is cauculated as:

$$\frac{Num(win)_{I.I.D}}{Num(all)} \tag{1}$$

$Num(win)_{I.I.D}$ refers to the number of samples where the performance of the I.I.D calibration set exceeds that of the OOD calibration set, while $Num(all)$ denotes the total number of data samples. The performance of the OOD calibration set is calculated as the average performance across all OOD calibration sets to eliminate discrepancies in the number of OOD calibration sets. The results are displayed in Tab. 3.

We are surprised to find that the calibration set using I.I.D datasets do not achieve better results in more than half of the settings that the winning rates are all near to 0.5, which is a rather counter-intuitive finding. Based on previous research on OOD generalization, using I.I.D data for pre-training or fine-tuning typically yields performance that far exceeds that of OOD data (Yuan et al., 2024). Additionally, experimental results (Wang et al., 2024; Albalak et al., 2024) indicate that fine-tuning large models using domain-relevant data yields better performance for specific downstream tasks. However, during the quantization phase, using datasets with the same or similar distribution as the test dataset did not significantly improve the performance of the quantized model.

**Guess**. *LLMs may not require highly relevant data related to downstream tasks to recover the performance loss due to quantization.* For the quantization task, the current design of quantization algorithms aims to restore the performance of full-precision models, rather than adapt to downstream tasks. Therefore, the model does not need data that closely resembles the downstream task, but rather a small amount of data to restore its capabilities, and it is not very sensitive to this subset of data. Thus, when the distribution difference between the calibration and test data is not significant, I.I.D data cannot yield better results. This also explains why the continuous increase of the calibration dataset leads to diminishing returns, rather than following the scaling laws like pre-training data (Williams & Aletras, 2023).

Table 3: The winning rate of I.I.D calibration data against OOD calibration data in three groups of experiments in S2.

| Distribution Shift | Winning Rate | Results |
| --- | --- | --- |
| Cross-dataset English Task | 0.45 | Tab. 2 |
| Cross-dataset Chinese Task | 0.61 | Tab. 13 |
| Cross-subject Chinese Task | 0.43 | Tab. 14 |

**Clue**. *We conduct a comparative experiment using C4 (Raffel et al., 2020) as the calibration set on BOSS and present the result in Tab 4*, which is a standard setting usually used for quantization. We observed that the performance of C4 is similarly close and random to that of I.I.D/OOD datasets, without a unified conclusion indicating which one consistently performs better. This result further validates our speculation that the choice of data during the quantization phase is more robust compared to other stages of data selection.

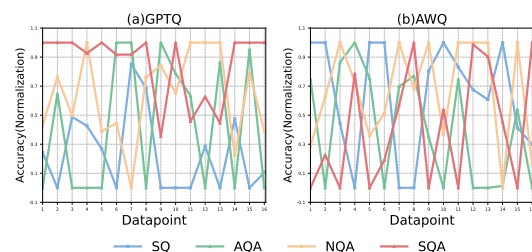

Figure 3: Normalized accuracy on EQA task using GPTQ and AWQ method. The left figure displays GPTQ results, while the right figure displays AWQ results.

Table 4: Results of C4 compared to I.I.D and OOD dataset as calibration set. We use GPTQ and test on the EQA task on BOSS. C4' is selected using a different random seed. The two best performances are denoted in descending order with red and orange respectively.

| Test | Gene. | W/A | Calib. | | | | | | Test | Gene. | W/A | Calib. | | | | | |
|------|-------|-----|--------|------|------|------|------|------|------|-------|-----|------|------|------|------|------|------|
| | | | C4 | C4' | SQ | AQA | NQA | SQA | | | | C4 | C4' | SQ | AQA | NQA | SQA |
| SQ | 0-shot | 4/16 | 54.50 | 51.71 | 53.84 | 52.73 | 54.69 | 57.31 | NQA | 0-shot | 4/16 | 39.79 | 38.98 | 37.94 | 38.76 | 38.63 | 38.23 |
| | | 3/16 | 54.29 | 54.73 | 45.31 | 48.86 | 49.49 | 50.79 | | | 3/16 | 35.79 | 36.68 | 31.36 | 33.79 | 33.37 | 34.45 |
| | 1-shot | 4/16 | 67.73 | 67.42 | 67.04 | 65.97 | 67.06 | 68.16 | | 1-shot | 4/16 | 48.80 | 49.00 | 48.55 | 49.30 | 49.73 | 49.09 |
| | | 3/16 | 63.72 | 64.64 | 60.76 | 58.84 | 63.34 | 63.01 | | | 3/16 | 45.66 | 45.03 | 44.38 | 43.35 | 46.95 | 45.61 |
| AQA | 0-shot | 4/16 | 30.80 | 27.61 | 28.00 | 27.12 | 28.40 | 30.40 | SQA | 0-shot | 4/16 | 46.29 | 46.03 | 42.58 | 45.72 | 46.21 | 44.20 |
| | | 3/16 | 25.04 | 27.91 | 21.81 | 25.28 | 23.35 | 24.99 | | | 3/16 | 31.13 | 33.38 | 30.19 | 26.99 | 28.49 | 33.73 |
| | 1- shot | 4/16 | 36.43 | 36.23 | 35.50 | 36.11 | 31.97 | 35.77 | | 1-shot | 4/16 | 62.48 | 61.44 | 56.04 | 61.89 | 60.92 | 62.17 |
| | | 3/16 | 33.45 | 34.01 | 31.39 | 29.54 | 31.60 | 32.24 | | | 3/16 | 53.25 | 53.00 | 43.46 | 42.83 | 45.17 | 48.82 |

**Interesting Findings.** *For a specific algorithm, there may exist one or more datasets that enhance the performance of the quantized model, independent of whether the dataset is I.I.D or OOD.* We present the results of normalizing the performance of the GPTQ and AWQ methods to the range [0, 1] on the EQA task in Fig. 3. The detailed normalization process is provided in Appendix C.5. It is evident that for GPTQ, the SQA dataset consistently exhibits good performance, while the SQ dataset consistently shows poor performance. However, this conclusion does not hold for AWQ, as all datasets demonstrate more random performance under AWQ. This may be related to the differing utilization of calibration data by algorithms.

## 4 MI-OPTIMIZE: LLM QUANTIZATION TOOLBOX

**Overview.** MI-optimize is a versatile tool designed for the quantization and evaluation of LLMs. The library's seamless integration of various quantization methods and evaluation techniques empowers users to customize their approaches according to specific requirements and constraints, providing a high level of flexibility. Although LLMs excel in various NLP tasks, their computational and memory demands may limit their deployment in real-time applications and on resource-constrained devices. MI-optimize addresses this challenge by employing quantization techniques to compress these models, ensuring they maintain performance while remaining adaptable to a wide range of scenarios. Fig. 4 illustrates the framework of MI-optimize, which comprises five main modules: the Configura-

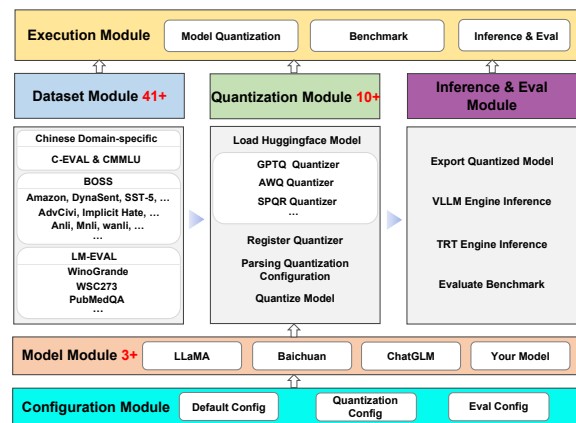

Figure 4: Overview of the Quantization and Evaluation Framework.

Table 5: Perplexity (PPL) of the LLaMA-2-7B model using SmoothQuant and a combination of SmoothQuant for activations and GPTQ for weight quantization on the WikiText-2 (Wiki2), Penn Treebank (PTB), and C4 datasets.

| Method | W/A | Wiki2 | C4 | PTB |
|---|---|---|---|---|
| Baseline | 16/16 | 5.47 | 37.92 | 7.22 |
| Smoothquant | 8/8 | 19.70 | 3026.75 | 11.27 |
| Smoothquant+GPTQ | 8/8 | 21.18 | 3110.05 | 11.27 |
| Smoothquant | 4/8 | 34.87 | 5133.82 | 20.82 |
| Smoothquant+GPTQ | 4/8 | **22.95** | **1359.59** | **13.39** |
| Smoothquant | 3/8 | 24041.06 | 42625.86 | 29585.39 |
| Smoothquant+GPTQ | 3/8 | **290.77** | - | **231.02** |

Table 6: Comparison with other quantization toolboxs.

| ToolBox | Number of Methods | Number of Benchmark&Datasets | Quantization Backend | Quantization Method Combination |
|---|---|---|---|---|
| Hugging Face Quanto library | 1 | - | ✓ | ✗ |
| qllm-eval (Li et al., 2024) | 3 | 30+ | ✗ | ✗ |
| TensorRT-LLM | 5 | - | ✓ | ✗ |
| VLLM (Kwon et al., 2023) | 4 | - | ✓ | ✗ |
| llama.cpp | 1 | - | ✓ | ✗ |
| Our Toolbox | **10+** | **40+** | ✓ | ✓ |

tion, Quantization, Evaluation, Inference,
and Execution modules. Tab. 6 presents the differences between our toolbox and other toolboxes. We provide a more comprehensive explanation of our toolbox in Appendix A.

**Experimental Setup and Results.** To validate the framework's capability of combining mixed quantization methods, we conduct experiments using the LLaMA-2-7B model (Touvron et al., 2023). We test the model using SmoothQuant and a combination of SmoothQuant for activations and GPTQ for weight quantization on WikiText-2 (Wiki2) (Merity et al., 2016), Penn Treebank (PTB) (Marcus et al., 1994), and C4 (Raffel et al., 2020) datasets, and measure the perplexity (PPL) of the quantized models. Quantization is implemented using PyTorch. All quantization experiments are exclusively conducted on the LLaMA-2-7B model, utilizing a single NVIDIA V100 GPU. For calibration, we utilize a dataset consisting of 128 random segments, each containing 512 tokens, extract from the C4 dataset. These segments represent generic text data, sourced from randomly crawled websites, ensuring that the quantization process does not rely on task-specific information. Our quantization setup employ SmoothQuant with default activation quantization of 8 bits. We utilize groupwise quantization with a group size of 128.

The results presented in Tab. 5 indicate several key findings. Comparing SmoothQuant with SmoothQuant + GPTQ configurations, it is evident that the latter consistently outperforms the former across all bit-width settings. This suggests that the combined use of SmoothQuant and GPTQ leads to a notable improvement in model performance. Particularly, at bit-widths of 4 and 3, the SmoothQuant + GPTQ method demonstrates a significant reduction in perplexity compared to SmoothQuant alone, indicating the pronounced effectiveness of GPTQ in reducing perplexity.

## 5 RELATED WORK

**Quantization of LLMs.** Quantization techniques for LLMs mainly include Post-Training Quantization (PTQ) and Quantization-Aware Training (QAT). PTQ does not require retraining the model and is typically suitable for situations with limited computational resources (Frantar et al., 2022; Dettmers et al., 2023; Lin et al., 2023; Xiao et al., 2023; Chee et al., 2024; Yao et al., 2022; Shao et al., 2023). QAT simulates the effects of quantization throughout the entire training process, enabling the model to adapt to low-precision representations during training, which typically leads to higher performance (Liu et al., 2023b; Dettmers et al., 2024). It's worth noting that in this paper, we consider applying quantization directly on the pretrained LLMs instead of performing quantization-aware finetuning for the quantized LLMs (such as variants of QLoRA (Dettmers et al., 2024; Yi et al., 2024; Xu et al., 2023)) because the latter typically needs the former for initialization.

Table 7: Comparison with related works.

| Work | Scenario | Number of Benchmark&Datasets |
|---|---|---|
| Jaiswal et al. (2023) | S1 | 5 |
| Williams & Aletras (2023) | S1 | 10 |
| Liu et al. (2023a) | S1 | 4 |
| Jin et al. (2024a) | S1 | 10 |
| Li et al. (2024) | S1 | 19 |
| Huang et al. (2024a) | S1 | 9 |
| Our Method | **S1&S2** | **40+** |

**Evaluation of quantized LLMs.** Numerous studies have undertaken evaluations of the performance of quantized LLMs (Frantar et al., 2022; Dettmers et al., 2023; Lin et al., 2023; Xiao et al., 2023; Chee et al., 2024; Jaiswal et al., 2023; Williams & Aletras, 2023; Li et al., 2024; Liu et al., 2023a; Jin et al., 2024b; Huang et al., 2024a). The majority of assessments employ fixed calibration set, primarily focusing on language modeling tasks (Raffel et al., 2020; Marcus et al., 1994; Merity et al., 2016) and standard NLP tasks (Zellers et al., 2019; Paperno et al., 2016; Tata & Patel, 2003; Clark et al., 2018; Sakaguchi et al., 2021; Mihaylov et al., 2018; Mostafazadeh et al., 2016). Certain investigations have deviated from the practice of using fixed calibration set, extending them to encompass a broader spectrum of crawled web text and pre-training data, while also conducting multiple random samplings for calibration set selection (Williams & Aletras, 2023). Additionally, certain studies have conducted assessments encompassing a broader array of downstream task types and datasets, approaching the evaluation from various angles (Liu et al., 2023a; Jaiswal et al., 2023; Li et al., 2024).

**Differences between our work and related work.** We have provided a detailed presentation of the differences between our work and related work in Tab. 7. The related work primarily addresses scenarios similar to S1 in our experiments and does not involve experiments related to the S2 scenario. Additionally, while some studies have considered distribution shifts (such as zero-shot and in-context learning), the scope of these shifts is limited, and the calibration datasets are fixed. This limitation results in a lack of systematic analysis regarding generalization capability and distribution shifts. These evaluations did not account for high-level generalization scenario classifications or assess variations in generalization ability across different settings. (Liu et al., 2023a) investigated the impact of quantization on model emergent abilities, evaluating OOD generalization tasks including zero-shot and in-context learning (e.g., ICL, CoT, instruction following). The evaluation types are similar to those in our S1 scenario but with limited scope. (Jaiswal et al., 2023) argued that perplexity (PPL) is not a good evaluation metric and thus evaluated numerous popular zero-shot tasks, similar to our S1 experiments but with limited scope and extent of distribution shifts. Li et al. (2024) evaluated a broader range of tasks and capabilities compared to previous work, but still focused on our S1 scenario and did not include tests for various distribution shifts.

# 6 FUTURE WORK AND CONCLUSION

Despite comprehensive evaluation on over 50 datasets, our study acknowledges the need for a more thorough assessment of models and quantization algorithms. Future work could involve a more extensive evaluation framework. Additionally, the developed toolbox does not yet support all quantization algorithms and large models. Further development is warranted to expand its capabilities.

We investigat the generalization ability of quantized LLMs, proposing two evaluation scenarios and testing them on our own implemented platform. S1 demonstrates that the quantized LLM maintains its generalization capability under all situations except those involving extreme bit quantization. Building upon this foundation, we introduce a novel generalization shift evaluation framework in S2, which investigates methods for enhancing generalization ability from a data perspective. Drawing from our evaluation results, we find that quantized models do not benefit from the alignment between calibration and test distributions. Further investigation revealed that this may be attributed to the fact that quantized models do not require a substantial amount of data relevant to downstream tasks to recover performance. Our work unveils the relationship between calibration data and test data, prompting the development of novel methods for optimizing calibration data collection, which is overlooked in the current field of model quantization. Lastly, we provided a modular and scalable toolbox to this topic to facilitate future research.

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

## A  MORE DETAILS OF OUR TOOLBOX

Fig. 4 illustrates the framework of MI-optimize, which comprises five main modules: the Configuration, Quant, Evaluation, Inference, and Execution modules. Combining these modules forms a cohesive pipeline that provides researchers with a reliable experimental environment, with each module responsible for a specific step in the pipeline. The subsequent sections will provide a detailed description of the implementation of each module.

- **Configuration Module**: Manages all parameters involved in the framework, including default settings, quantization configurations, and evaluation configurations.
- **Model Module**: Contains various pre-trained models such as LLaMA (Touvron et al., 2023), Baichuan (Yang et al., 2023), ChatGLM (Du et al., 2021), and custom user models.
- **Dataset Module**: Handles different datasets, including Chinese domain-specific datasets (e.g., C-EVAL (Huang et al., 2024b) and CMMLU (Li et al., 2023)), the BOSS benchmark (Yuan et al., 2024), general datasets (e.g., Amazon reviews, Dynasent), and LM-EVAL datasets (e.g., Winogrande, WSC273).
- **Quant Module**: Responsible for loading pre-trained models, applying various quantization methods (e.g., GPTQ (Frantar et al., 2022), AWQ (Lin et al., 2023), SPQR (Dettmers et al., 2023)), and performing the actual model quantization.
- **Inference & Eval Module**: Exports the quantized model, runs inference using engines such as VLLM (Kwon et al., 2023) and TensorRT, and evaluates benchmark performance.
- **Execution Module**: Oversees the primary tasks of model quantization, benchmarking, and the combined process of quantization and evaluation.

**Key Features Supported by MI-optimize.**

- Quantization of LLMs to reduce computational and memory requirements: MI-optimize focuses on reducing the computational and memory footprint of large language models through advanced quantization techniques, making them more suitable for deployment in resource-limited environments.
- Support for various quantization algorithms: The framework supports a wide range of quantization algorithms, including RTN, GPTQ Frantar et al. (2022), AWQ Lin et al. (2023), SpQR Dettmers et al. (2023), ZeroQuant Yao et al. (2022), SmoothQuant Xiao et al. (2023), QuIP Chee et al. (2024), and FP8. This flexibility allows users to choose the most appropriate method for their specific use case, optimizing performance and resource usage.
- Evaluation on OOD tasks using benchmarks: MI-optimize includes tools for evaluating quantized models on out-of-distribution (OOD) tasks using established benchmarks such as BOSS. This ensures that the models maintain their performance even when encountering data that differs from their training set.
- Support for multiple datasets: The framework supports multiple datasets for both calibration and testing purposes. Users can also incorporate custom datasets to better align the model's performance with their specific requirements.
- Command-line interface for easy integration and automation: MI-optimize provides a command-line interface that facilitates easy integration into existing workflows and automation of the quantization and evaluation processes, streamlining the deployment pipeline.
- Support for combination of quantization methods: The framework allows for the combination of different quantization methods within the same model. Different layers can apply different quantization algorithms, and even multiple quantization algorithms can be applied to the same layer. This granular control helps optimize model performance and efficiency.
- Ease of adding new quantization algorithms: Researchers can easily add new quantization algorithms to the MI-optimize repository. This modularity ensures that the framework remains up-to-date with the latest advancements in quantization techniques.
- Customer tools for model quantization and evaluation: Customers can install the tools provided by MI-optimize to quantize and evaluate their own models. This empowers users to tailor the framework to their specific needs, ensuring optimal model performance in their applications.

## B DATASETS

In this section, we present all the datasets utilized in the experiments, encompassing their evaluated tasks and abilities, assessment metrics, and dataset sizes. Tab. 8 and 9 provide a comprehensive summary of all the datasets.

### B.1 DATASETS IN S1

**Common sense reasoning. WinoGrande** (Sakaguchi et al., 2021) is a large-scale coreference resolution task dataset derived from extensive internet text, aimed at addressing ambiguous and complex coreference relationships. **WSC273** (Levesque et al., 2012) comprises 273 coreference resolution problems derived from the classic Winograd Schema Challenge, primarily assessing the common-sense reasoning capabilities of natural language understanding systems. **GLUE-WNLI** (Wang et al., 2018) is designed to test coreference resolution capability, which involves determining which noun a pronoun in a sentence refers to. It is sourced from the Winograd Schema Challenge. **HellaSwag** (Zellers et al., 2019) is generated from web videos and Wikipedia articles and is used to infer the most suitable continuation for text segments in multiple-choice tasks. **SWAG** (Zellers et al., 2018) is generated based on video descriptions, aiming to predict plausible subsequent scenarios for video events. **PIQA** (Tata & Patel, 2003) is a dataset for reasoning about physical common sense, derived from physics problems and solutions, designed to evaluate algorithms' reasoning abilities in physical environments.

**Mathematical reasoning. MathQA** (Amini et al., 2019) is collected from the MathQA website, consisting of 37,200 mathematical questions, with the task being to automatically answer mathematical questions.

**Multi-turn dialogue reasoning. MuTual** (Cui et al., 2020) and **Mutual_plus** (Cui et al., 2020) is a retrieval-based dataset for multi-turn dialogue reasoning, which is modified from Chinese high school English listening comprehension test data.

**Bias diagnosis and mitigation. CrowS-Pairs** (Nangia et al., 2020) is derived from a wide range of internet text and is designed to evaluate social biases in language models. **Toxigen** (Hartvigsen et al., 2022) is for implicit hate speech detection.

**Scientific knowledge question answering. PubMedQA** (Jin et al., 2019) is a biomedical question answering dataset sourced from PubMed articles, aimed at evaluating systems' understanding and answering capabilities of biomedical texts. **OpenBookQA** (Mihaylov et al., 2018) is a new kind of question-answering dataset modeled after open book exams for assessing human understanding of a subject. It originates from open science education resources. **SciQ** (Welbl et al., 2017) is a high-quality, science-themed multiple-choice dataset constructed manually. **ARC-Easy** (Clark et al., 2018) originates from science exams administered in American elementary through high schools, assessing fundamental scientific knowledge. **ARC-Challenge** (Clark et al., 2018) presents challenging scientific questions aimed at testing higher-level scientific comprehension and reasoning abilities. **MC-TACO** (Zhou et al., 2019) consists of temporal common-sense questions sourced from a wide range of internet texts, designed for temporal common-sense reasoning tasks.

**Reading comprehension. RACE** (Lai et al., 2017) is a large-scale reading comprehension dataset sourced from English exams for Chinese middle school and high school students, aimed at testing reading comprehension abilities. **QA4MRE** (Peñas et al., 2013) is created for the CLEF 2011/2012/2013 shared tasks, aimed at testing cross-domain reading comprehension abilities.

**Natural language inference. GLUE-MNLI** (Wang et al., 2018) is a natural language inference dataset comprising pairs of sentences sourced from various text genres such as novels, telephone conversations, and news articles. **GLUE-MNLI-Mismatched** (Wang et al., 2018) is utilized to evaluate the generalization capability of models on unseen text genres, with sentence pairs sourced from the same origins as GLUE-MNLI. **GLUE-RTE** (Wang et al., 2018) is sourced from news reports and Wikipedia. **GLUE-QNLI** (Wang et al., 2018) originates from the Stanford University's SQuAD dataset. **ANLI** (Nie et al., 2019) is a large-scale adversarial natural language inference dataset divided into three difficulty levels. It is constructed by employing adversarial search techniques to generate challenging questions based on human annotations.

**Sentiment analysis. GLUE-SST** (Wang et al., 2018) is sourced from movie reviews, and its task involves sentiment classification, which entails determining the emotional inclination of a sentence.

**Syntax phenomena evaluation. BLiMP** (Warstadt et al., 2020) is a challenge set for evaluating what language models know about major grammatical phenomena in English. BLiMP consists of 67 sub-datasets, each containing 1000 minimal pairs isolating specific contrasts in syntax, morphology, or semantics. The data is automatically generated according to expert-crafted grammars.

## B.2   DATASETS IN S2

**Extractive question answering in BOSS. SQuAD** (Rajpurkar et al., 2016) is a collection of question-answer pairs derived from Wikipedia articles. **AdversarialQA** (Bartolo et al., 2020) formulates adversarial questions within the SQuAD context, utilizing a collaborative process involving both human annotators and models. **NewsQA** (Trischler et al., 2016) crafts questions based on CNN news articles, each demanding reasoning for answers, rather than relying solely on lexical overlap and textual entailment. **SearchQA** (Dunn et al., 2017) employs a reverse construction approach, utilizing the Google search engine to fetch pertinent contexts for each question-answer pair from the J!Archive website.

**Sentiment analysis in BOSS. Amazon** (McAuley & Leskovec, 2013) is a dataset comprising reviews across 29 distinct product categories from the Amazon website. **DynaSent** (Potts et al., 2020) constructs a dataset by identifying challenging sentences from existing collections and generating adversarial counterparts through human-and-model collaborative annotation. **SemEval** (Nakov et al., 2019) offers a three-class sentiment analysis dataset centered on Twitter content. **SST** (Socher et al., 2013) features sentence-level movie reviews sourced from the Rotten Tomatoes website.

**Natural language inference in BOSS. MNLI** (Williams et al., 2017) offers sentence pairs across ten diverse categories of written and verbal communication, showcasing various styles, topics, and formalities. **ANLI** (Nie et al., 2019) is an adversarial dataset created using a human-and-model-in-the-loop method, featuring premises primarily sourced from Wikipedia and hypotheses crafted by human adversaries. **ContractNLI** (Koreeda & Manning, 2021) treats individual contracts as premises and applies a consistent set of hypotheses across the dataset. **WANLI** (Liu et al., 2022) is generated by GPT-3, containing examples that include challenging patterns initially identified in MNLI.

**Toxic detection in BOSS. Civil Comments** (Borkan et al., 2019) features public comments from the Civil Comments platform, encompassing a diverse user base and various subtypes of toxic text. **AdvCivil** introduces a new toxic dataset, derived from Civil Comments through textual adversarial attacks within an automated model-in-the-loop adversarial pipeline. **Implicit Hate** (ElSherief et al., 2021) includes toxic tweets that are both explicit and implicit, with the latter capable of evading keyword-based toxic detection systems. **ToxiGen** (Hartvigsen et al., 2022) is generated by GPT-3 and contains subtly and implicitly toxic texts targeting 13 minority groups.

**Chinese domain-specific. C-Eval** (Huang et al., 2024b) is a comprehensive Chinese evaluation suite for foundation models. It consists of 13948 multi-choice questions spanning 52 diverse disciplines and four difficulty levels, primarily encompassing humanities, social sciences, STEM, and other 4 categories. **CMMLU** (Li et al., 2023) is a comprehensive Chinese evaluation benchmark designed specifically to assess language models' knowledge and reasoning abilities within Chinese contexts. CMMLU covers 67 topics ranging from fundamental subjects to advanced professional levels. It encompasses topics such as STEM requiring calculation and reasoning, humanities and social sciences necessitating knowledge, and everyday knowledge such as Chinese driving rules.

## C   EXPERIMENT DETAILS

In this section, we will present all the details of our experiment, including hardware resources, experimental setup, hyperparameter selection, and data selection. Besides, Our benchmark suite is available in the supplementary materials.

Table 8: Summary of the datasets in S1.

| Scenario | Task&Ability | Dataset | Gene. | Metric | Size |
|---|---|---|---|---|---|
| S1 | Common sense reasoning | WinoGrande Sakaguchi et al. (2021) | 0/5 | Acc | 1267 |
| S1 | Common sense reasoning | WSC273 Levesque et al. (2012) | 0/5 | Acc | 273 |
| S1 | Common sense reasoning | GLUE-WNLI Wang et al. (2018) | 0/5 | Acc | 71 |
| S1 | Common sense reasoning | HellaSwag Zellers et al. (2019) | 0/5 | Acc | 10042 |
| S1 | Common sense reasoning | SWAG Zellers et al. (2018) | 0/5 | Acc | 20006 |
| S1 | Common sense reasoning | PIQA Tata & Patel (2003) | 0/5 | Acc | 1838 |
| S1 | Mathematical reasoning | MathQA Amini et al. (2019) | 0/5 | Acc | 2985 |
| S1 | Multi-turn dialogue reasoning | Mutual Cui et al. (2020) | 0/5 | R2 | 886 |
| S1 | Multi-turn dialogue reasoning | Mutual_Plus Cui et al. (2020) | 0/5 | R2 | 886 |
| S1 | Bias diagnosis and mitigation | CrowS-Pairs Nangia et al. (2020) | 0 | Pct_stereotype | 6708 |
| S1 | Bias diagnosis and mitigation | Toxigen Hartvigsen et al. (2022) | 0/5 | Acc | 940 |
| S1 | Scientific knowledge question answering | PubMedQA Jin et al. (2019) | 0/5 | Acc | 1000 |
| S1 | Scientific knowledge question answering | OpenBookQA Mihaylov et al. (2018) | 0/5 | Acc | 500 |
| S1 | Scientific knowledge question answering | SciQ Welbl et al. (2017) | 0/5 | Acc | 1000 |
| S1 | Scientific knowledge question answering | ARC-Easy Clark et al. (2018) | 0/5 | Acc | 2376 |
| S1 | Scientific knowledge question answering | ARC-Challenge Clark et al. (2018) | 0/5 | Acc | 1172 |
| S1 | Scientific knowledge question answering | MC-TACO Zhou et al. (2019) | 0/5 | F1 | 9442 |
| S1 | Reading comprehension | RACE Lai et al. (2017) | 0/5 | Acc | 1045 |
| S1 | Reading comprehension | QA4MRE Peñas et al. (2013) | 0/5 | Acc | 564 |
| S1 | Natural language inference | GLUE-MNLI Wang et al. (2018) | 0/5 | Acc | 9815 |
| S1 | Natural language inference | GLUE-MNLI-Mismatched Wang et al. (2018) | 0/5 | Acc | 9832 |
| S1 | Natural language inference | GLUE-RTE Wang et al. (2018) | 0/5 | Acc | 277 |
| S1 | Natural language inference | GLUE-QNLI Wang et al. (2018) | 0/5 | Acc | 5463 |
| S1 | Natural language inference | ANLI Nie et al. (2019) | 0/5 | Acc | 3200 |
| S1 | Sentiment analysis | GLUE-SST Wang et al. (2018) | 0/5 | Acc | 872 |
| S1 | Syntax phenomena evaluation | BLiMP Warstadt et al. (2020) | 5 | Acc | 67000 |

Table 9: Summary of the datasets in S2.

| Scenario | Task&Ability | Dataset | Gene. | Metric | Size |
|---|---|---|---|---|---|
| S2 | Extractive question answering | SQuAD Rajpurkar et al. (2016) | 0/1 | F1 | 10570 |
| S2 | Extractive question answering | AdversarialQA Bartolo et al. (2020) | 0/1 | F1 | 2694 |
| S2 | Extractive question answering | NewsQA Trischler et al. (2016) | 0/1 | F1 | 3912 |
| S2 | Extractive question answering | SearchQA Dunn et al. (2017) | 0/1 | F1 | 16680 |
| S2 | Sentiment analysis | Amazon McAuley & Leskovec (2013) | 0/3 | Acc | 38905 |
| S2 | Sentiment analysis | DynaSent Potts et al. (2020) | 0/3 | Acc | 4020 |
| S2 | Sentiment analysis | SemEval Nakov et al. (2019) | 0/3 | Acc | 20322 |
| S2 | Sentiment analysis | SST Socher et al. (2013) | 0/3 | Acc | 767 |
| S2 | Natural language inferenc | MNLI Williams et al. (2017) | 0/3 | Acc | 9815 |
| S2 | Natural language inferenc | ANLI Nie et al. (2019) | 0/3 | Acc | 2900 |
| S2 | Natural language inferenc | ContractNLI Koreeda & Manning (2021) | 0/3 | Acc | 1791 |
| S2 | Natural language inferenc | WANLI Liu et al. (2022) | 0/3 | Acc | 4700 |
| S2 | Toxic detection | Civil Comments Borkan et al. (2019) | 0/2 | Acc | 97320 |
| S2 | Toxic detection | AdvCivil | 0/2 | Acc | 523 |
| S2 | Toxic detection | Implicit Hate ElSherief et al. (2021) | 0/2 | Acc | 21180 |
| S2 | Toxic detection | ToxiGen Hartvigsen et al. (2022) | 0/2 | Acc | 641 |
| S2 | Chinese domainspecific | CEVAL Huang et al. (2024b) | 0/5 | Acc | 13948 |
| S2 | Chinese domainspecific | CMMLU Li et al. (2023) | 0/5 | Acc | 11917 |

## C.1 HARDWARE RESOURCES

In our experiments, we utilize one computer with 8 AMD Aldebaran GPUs and two computers with 2 NVIDIA Tesla V100 GPUs each. Specifically, each AMD Aldebaran GPU has 64GB of memory, totaling 512GB. Each NVIDIA Tesla V100 GPU has 32GB of memory, totaling 128GB.

## C.2 EXPERIMENT DETAILS IN S1

**Experimental Setup.** We quantize LLaMA2-7B (Touvron et al., 2023) using the GPTQ (Frantar et al., 2022), SpQR (Dettmers et al., 2023) methods. We quantize the weights to 2-4 bits and test 16 bits as reference. The quantization is implemented using our custom toolbox, maintaining consistency with the original method in all experimental details.

**Hyperparameter Selection.** For the GPTQ (Frantar et al., 2022) method, we set the group-size parameter to 128 and apply block-sequential as well as layer-sequential quantization. For the SpQR (Dettmers et al., 2023) method, we set the group-size parameter to 128 and apply block-sequential quantization. Throughout the quantization process, we use 128 calibration examples. In the few-shot setting, the number of selected examples corresponds to LM Evaluation Harness (Gao et al., 2021b), remaining at 5-shot.

**Data Selection.** We follow GPTQ (Frantar et al., 2022) and randomly sample 128 samples from C4-en-val (Raffel et al., 2020) as the calibration set with a random seed of 42. For the selection of test data, we use the test splits of ANLI (Nie et al., 2019), ARC (Clark et al., 2018), CrowS-Pairs (Nangia et al., 2020), GLUE-MNLI-Mismatched (Wang et al., 2018), MathQA (Amini et al., 2019), MCTACO (Zhou et al., 2019), OpenBookQA (Mihaylov et al., 2018), RACE (Lai et al., 2017), SciQ (Welbl et al., 2017), Toxigen (Hartvigsen et al., 2022), and WSC273 (Levesque et al., 2012) as the test set. We use the validation splits of GLUE-SST, GLUE-MNLI, GLUE-QNLI, GLUE-WNLI, GLUE-RTE (Wang et al., 2018), HellaSwag (Zellers et al., 2019), Mutual (Cui et al., 2020), PIQA (Tata & Patel, 2003), SWAG (Zellers et al., 2018), WinoGrande (Sakaguchi et al., 2021) as the test set. Additionally, we use the train splits of BLiMP (Warstadt et al., 2020), PubMedQA (Jin et al., 2019), and QA4MRE (Peñas et al., 2013) as the test set. For the selection of examples in the few-shot setting, we use the default setting.

## C.3 EXPERIMENT DETAILS IN S2

### C.3.1 BOSS

**Experimental Setup.** We quantize LLaMA2-7B (Touvron et al., 2023) using the GPTQ (Frantar et al., 2022), SpQR (Dettmers et al., 2023), AWQ (Lin et al., 2023), and SmoothQuant (Xiao et al., 2023) methods. We quantize the weights to 3-4 bits, and for smoothquant, we further quantize the activations to 8 bits. The quantization is implemented using our custom toolbox, maintaining consistency with the original method in all experimental details.

**Hyperparameter Selection.** For the GPTQ (Frantar et al., 2022) method, we set the group-size parameter to 128 and apply block-sequential as well as layer-sequential quantization. For the SpQR (Dettmers et al., 2023) method, we set the group-size parameter to 128 and apply block-sequential quantization. For the AWQ (Lin et al., 2023) method, we set the group-size parameter to 128. Throughout the quantization process, we use 128 calibration examples. In the few-shot setting, the number of selected examples corresponds to those in BOSS. Specifically, EQA is 1-shot, SA and NLI are 3-shot, and TD is 2-shot. The prompt template is presented in Tab. 10.

**Data Selection.** For the calibration set, we use 128 calibration examples. For SQuAD (Rajpurkar et al., 2016) dataset in EQA, Amazon (McAuley & Leskovec, 2013) dataset in SA, MNLI (Williams et al., 2017) dataset in NLI, and Civil Comments (Borkan et al., 2019) dataset in TD, as the original datasets include train and test splits, we directly select the first 128 instances from the train split as the calibration set. For the remaining datasets, given that the original datasets exclusively contain a test split, we randomly sample 300 instances from the test split to form a train split, subsequently removing the sampled data from the test split. We use the first 128 instances from the sampled train split as the calibration set. The random seed is set to 42. The code for processing the original BOSS benchmark will be placed in our GitHub repository. Concerning the selection of examples in the few-shot setting, we maintain consistency with BOSS. For datasets lacking examples, we

appropriately select suitable samples from the portion of the train split not chosen as part of the calibration set. For the test data, we use the test split of each dataset as the testing dataset.

### C.3.2 Chinese domain-specific

**Experimental Setup.** We quantize Baichuan2-7B-Base (Yang et al., 2023) using the GPTQ (Frantar et al., 2022), SpQR (Dettmers et al., 2023), AWQ (Lin et al., 2023), and Smoothquant (Xiao et al., 2023) methods. We quantize the weights to 3-4 bits, and for smoothquant, we further quantize the activations to 8 bits. The quantization is implemented using our custom toolbox, maintaining consistency with the original method in all experimental details. Since the test split of C-EVAL was not publicly available, we upload the test answers to the official platform to obtain the results.

**Hyperparameter Selection.** For the GPTQ (Frantar et al., 2022) method, we set the group-size parameter to 128 and apply block-sequential as well as layer-sequential quantization. For the SpQR (Dettmers et al., 2023) method, we set the group-size parameter to 128 and apply block-sequential quantization. For the AWQ (Lin et al., 2023) method, we set the group-size parameter to 128. Throughout the quantization process, we use 128 calibration examples. In the few-shot setting, the number of selected examples corresponds to those in C-EVAL (Huang et al., 2024b) and CMMLU (Li et al., 2023), remaining at 5-shot.

**Data Selection.** For the calibration set, we use 128 calibration examples. For C-EVAL (Huang et al., 2024b), we utilize its validation split as the calibration set. For CMMLU (Li et al., 2023), we randomly select 300 instances from its test split for the train split, subsequently removing the sampled data from the test split. We use the first 128 instances from the sampled train split as the calibration set. The random seed is set to 42. As for the selection of examples in the few-shot setting, we remain consistent with the official standards of C-EVAL and CMMLU. The prompt template is presented in Tab. 10.

### C.4 Experiment Details of Comparision between C4 and I.I.D/OOD Calibration Set

**Experimental Setup.** We quantize LLaMA2-7B (Touvron et al., 2023) using the GPTQ (Frantar et al., 2022) method and test on the EQA task in BOSS. We quantize the weights to 3-4 bits. The quantization is implemented using our custom toolbox, maintaining consistency with the original method in all experimental details.

**Hyperparameter Selection.** For the GPTQ (Frantar et al., 2022) method, we set the group-size parameter to 128 and apply block-sequential as well as layer-sequential quantization. Throughout the quantization process, we use 128 calibration examples. In the few-shot setting, EQA task is 1-shot. The prompt template is presented in Tab. 10.

**Data Selection.** Regarding the data selection from C4 (Raffel et al., 2020) as the calibration set, We follow GPTQ (Frantar et al., 2022) and randomly sample 128 samples from C4-en-val as the calibration set with the random seeds of 42 and 567. Regarding the data selection from BOSS Yuan et al. (2024) as the calibration set, we remain consistent with the previous experiments in sec. C.3.1 and set the random seed to 42. For the test data, we use the test split of each dataset as the testing dataset.

### C.5 Experiment Details of Normalization

**Experimental Setup.** To eliminate performance differences between different downstream tasks and situations and better evaluate the quality of different datasets as calibration datasets, we normalize the results based on the same method, generation scenario, bits and different test sets, specifically the 1/4 row in Tab 2. We used Min-max normalization, calculated as follows:

$$x' = \frac{x - X_{min}}{X_{max} - X_{min}} \tag{2}$$

The normalized results range from $[0, 1]$. This allows for better comparison of the performance of different calibration sets and enables visualization of the performance differences between calibration sets, rather than just their rankings.

Table 10: Prompts for BOSS and Chinese domain-specific tasks. We maintain consistency with the official template provided by BOSS Yuan et al. (2024) and C-EVAL Huang et al. (2024b).

| Task | Prompt |
|---|---|
| EQA | ### Instruction ### 
 Solve the extractive question answering task. Refering to the passage below and extract answer for the question. The answer should be the shortest phrase as it can be. 
 ### Format ### 
 Passage: {{Passage}} // Question: {{Question}} // Answer: {{Answer}}. 
 ### Input ### 
 Passage: {{input_1}} // Question: {{input_2}} // Answer: |
| SA | ### Instruction ### 
 Solve the sentiment analysis task. Options for sentiment: negative, positive, neutral. 
 ### Format ### 
 Text: {{Text}} // Prediction: {{Prediction}} 
 ### Input ### 
 Text: {{input}} // Prediction: |
| NLI | ### Instruction ### 
 Solve the NLI task. Options for entailment relationship: entailment, neutral, contradiction. 
 ### Format ### 
 Premise: {{Premise}} // Hypothesis: {{Hypothesis}} // Prediction: {{Prediction}} 
 ### Input ### 
 Premise: {{input_1}} // Hypothesis: {{input_2}} // Prediction: |
| TD | ### Instruction ### 
 Solve the toxic detection task. Options for toxicity: benign, toxic. 
 ### Format ### 
 Text: {{Text}} // Prediction: {{Prediction}} 
 ### Input ### 
 Text: {{input}} // Prediction: |
| CDS | 以下是中国考试的单项选择题，请选出其中的正确答案。 |

# D    MORE EXPERIMENTS

## D.1    THE ROBUSTNESS OF DATA SELECTION WITH RESPECT TO RANDOM SEED

In the experiments conducted in the main text, we employ a random seed for the selection of train split and calibration set. In this section, we will alter the random seed to observe the sensitivity of the experiments to the random seed.

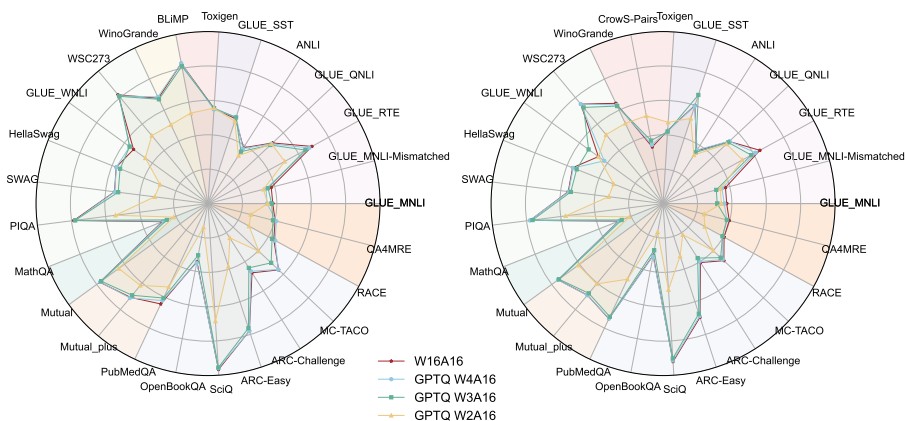

Figure 5: **S1**: evaluation of quantized LLaMA2-7B on several standard datasets. Quantization methods include GPTQ. Quantization bits include W4A16, W3A16, and W2A16, with W16A16 used as reference. The left figure shows 5-shot results, while the right figure shows 0-shot results. Different background colors represent different task types. The random seed is 42.

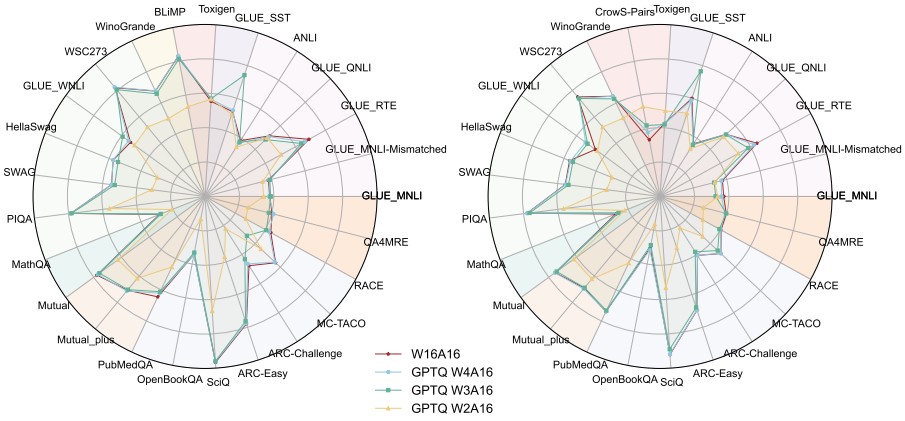

Figure 6: **S1**: evaluation of quantized LLaMA2-7B retested on several standard datasets. Quantization methods include GPTQ. Quantization bits include W4A16, W3A16, and W2A16, with W16A16 used as reference. The left figure shows 5-shot results, while the right figure shows 0-shot results. Different background colors represent different task types. The random seed is 567.

In S1, we randomly sampled 128 samples from c4-en-val as the calibration set and set the random seed to 42. We then modify the random seed to 567 and retest the GPTQ (Frantar et al., 2022) method. The results are presented in Fig. 5 and 6. We observe that the vast majority of datasets exhibited strong robustness to the selection of the calibration set, with performance trends remaining nearly identical across different random seeds. In the comparisons presented in Fig. 5 and Fig. 6, we can still observe that different task types exhibit varying sensitivities to quantization, and this conclusion remains consistent across different random seeds. We see that task types such as scientific knowledge QA,

reading comprehension, common sense reasoning, mathematical reasoning and sentiment analysis are highly sensitive to quantization, while task types like natural language inference demonstrate robustness against quantization.

In Cross-dataset distribution shift evaluation on BOSS in S2, we randomly sample some examples from the test split as the train split and use them as the calibration set, setting the random seed to 42. We modify the random seed to 567 and retest the SA and NLI experiments using GPTQ (Frantar et al., 2022) method. We present the average results with random seeds 42 and 567 in Tab. 11. The results indicate a certain robustness of the distribution shift experiment on BOSS towards the selection of the calibration set. For SA task, performance remains consistently better when using Amazon (McAuley & Leskovec, 2013) as the calibration set across different random seeds, and using SemEval (Nakov et al., 2019) as the calibration set performs better in most cases. However, the performance has consistently been poor when using DynaSent (Potts et al., 2020) as the calibration set. For NLI task, performance remains consistently better when using MNLI (Williams et al., 2017) as the calibration set across different random seeds.

Table 11: Cross-dataset distribution shift evaluation retested on Boss. The result represents the average values obtained with random seeds 42 and 567. "Calib." represents the calibration dataset, and "Gene." represents generalization scenario. To save space, abbreviations are used for datasets. Each row presents experimental results using different datasets as calibration sets on the same test dataset. The higher the metric, the better the performance. The two best performances are denoted in descending order with red and orange respectively. Note: Some datasets could not be used as calibration sets due to insufficient memory resources.

| Method | Test | Gene. | W/A | AZ | DS | SE | SST | Test | Gene. | W/A | MN | AN | WN | CN |
|---|---|---|---|---|---|---|---|---|---|---|---|---|---|---|
| GPTQ | AZ | 0-shot | 4/16 | 65.84 | 46.90 | 66.49 | 53.61 | MN | 0-shot | 4/16 | 0.25 | 0.31 | 0.25 | - |
| | | | 3/16 | 19.14 | 0.50 | 21.41 | 0.03 | | | 3/16 | 0.03 | 0.00 | 0.00 | - |
| | | 3-shot | 4/16 | 78.47 | 70.35 | 81.43 | 80.32 | | 3-shot | 4/16 | 43.28 | 34.18 | 41.46 | - |
| | | | 3/16 | 80.73 | 41.28 | 70.79 | 70.23 | | | 3/16 | 32.95 | 33.02 | 32.01 | - |
| | DS | 0-shot | 4/16 | 41.85 | 30.55 | 40.89 | 25.15 | AN | 0-shot | 4/16 | 0.74 | 0.57 | 0.74 | - |
| | | | 3/16 | 8.80 | 1.17 | 10.57 | 0.00 | | | 3/16 | 2.26 | 0.00 | 0.00 | - |
| | | 3-shot | 4/16 | 53.88 | 45.50 | 54.15 | 52.38 | | 3-shot | 4/16 | 34.1 | 33.52 | 33.76 | - |
| | | | 3/16 | 53.86 | 40.25 | 44.26 | 48.91 | | | 3/16 | 32.25 | 33.33 | 34.19 | - |
| | SE | 0-shot | 4/16 | 19.97 | 14.07 | 22.08 | 14.27 | WN | 0-shot | 4/16 | 0.09 | 0.08 | 0.10 | - |
| | | | 3/16 | 2.48 | 0.10 | 8.25 | 0.02 | | | 3/16 | 0.27 | 0.00 | 0.00 | - |
| | | 3-shot | 4/16 | 41.09 | 36.48 | 43.41 | 44.05 | | 3-shot | 4/16 | 42.16 | 42.15 | 39.925 | - |
| | | | 3/16 | 42.69 | 27.98 | 38.57 | 36.48 | | | 3/16 | 43.16 | 43.36 | 46.97 | - |
| | SST | 0-shot | 4/16 | 44.13 | 33.505 | 37.16 | 25.56 | CN | 0-shot | 4/16 | 0.03 | 0.50 | 0.00 | - |
| | | | 3/16 | 3.93 | 0.52 | 5.09 | 0.00 | | | 3/16 | 0.03 | 0.56 | 0.73 | - |
| | | 3-shot | 4/16 | 54.83 | 44.01 | 52.61 | 48.11 | | 3-shot | 4/16 | 35.93 | 36.67 | 32.27 | - |
| | | | 3/16 | 57.17 | 44.33 | 46.68 | 52.29 | | | 3/16 | 28.28 | 20.41 | 26.13 | - |

## D.2 SUPPLEMENTARY RESULTS OF S1

In this subsection, we present supplementary results in S1. Fig. 2 primarily illustrates the performance of each dataset under different quantization situations. Based on this, we categorized these datasets into 9 types of downstream tasks and calculated the average accuracy decline for each downstream task type under various quantization situations in Tab. 12. We can clearly observe that different downstream task types exhibit varying sensitivities to quantization. Scientific knowledge QA, reading comprehension, common sense reasoning, and mathematical reasoning are highly sensitive to quantization, with low-bit quantization leading to significant performance drops. For example, in the case of scientific knowledge QA and reading comprehension, the performance decline can reach up to 40%, while for common sense reasoning and mathematical reasoning, it may drop by around 25%. Notably, although sentiment analysis task also demonstrate high sensitivity to quantization, low-bit quantization can lead to substantial performance improvements. In contrast, tasks like natural language inference and multi-turn dialogue readoning are less sensitive to quantization, showing minimal performance variation across different methods and bit-widths, typically not exceeding 10%.

The varying sensitivity to quantization across different task types may be attributed to differences in full-precision performance. For tasks such as Scientific Knowledge QA, which can achieve a

Table 12: The specific percentages of performance degradation after quantization for each task type in S1. Performance degradation that are significantly high or low for all task types are marked in red red and blue blue, respectively

| Gene. | Method& Bits | Scientific knowledge QA | Reading comprehension | Natural language inference | Sentiment analysis | Bias diagnosis mitigation | Syntax phenomena evaluation | Common sense reasoning | Mathematical reasoning | Multi-turn dialogue reasoning |
|---|---|---|---|---|---|---|---|---|---|---|
| 0-shot | GPTQ-4bit | 0.68 | 3.09 | 2.36 | 0.76 | 1.18 | - | 1.93 | 1.44 | -0.16 |
| | GPTQ-3bit | 4.958 | 3.09 | 8.64 | -10.71 | 3.60 | - | -0.01 | 6.98 | 1.35 |
| | GPTQ-2bit | 40.60 | 34.81 | 9.33 | 13.19 | 8.23 | - | 27.57 | 24.22 | 15.71 |
| | SPQR-4bit | 0.83 | 0.55 | 2.36 | 4.02 | -0.02 | - | 0.35 | -0.12 | 0.10 |
| | SPQR-3bit | 1.22 | 3.11 | 2.36 | -8.99 | 1.89 | - | -1.26 | 7.11 | 0.51 |
| | SPQR-2bit | 5.25 | 4.18 | 5.56 | 4.21 | 1.05 | - | -0.39 | 4.33 | 1.37 |
| 5-shot | GPTQ-4bit | 0.87 | -1.98 | 4.54 | -2.21 | 1.15 | 0.00 | -0.67 | 2.81 | -0.17 |
| | GPTQ-3bit | 6.37 | 2.87 | 4.72 | -1.55 | -0.38 | 2.09 | 1.06 | 9.85 | 1.77 |
| | GPTQ-2bit | 39.11 | 36.81 | 12.42 | 2.88 | -0.38 | 35.51 | 33.36 | 25.09 | 15.09 |
| | SPQR-4bit | 1.65 | -0.55 | 1.23 | -0.89 | 0.38 | 0.47 | -0.72 | 1.17 | -0.40 |
| | SPQR-3bit | 2.75 | 1.87 | 4.47 | 1.11 | 0.77 | 1.05 | 3.33 | 1.06 | -0.13 |
| | SPQR-2bit | 5.72 | 4.40 | 3.87 | -22.79 | 0.58 | 1.02 | 1.90 | 6.45 | 1.56 |

maximum accuracy of up to 90%, extreme quantization results in a significant drop in performance, leading to a marked decrease in relative performance. In contrast, for natural language inference tasks with generally lower full-precision performance of around 30% to 40%, the models may not even meet the threshold for effectively solving natural language inference tasks, leaving little room for performance degradation under extreme quantization.

### D.3 FULL RESULTS OF CHINESE CROSS-DATASET AND CROSS-SUBJECT TRANSFER TASKS

In this subsection, we present all the results for the Chinese domain-specific tasks in S2 in sec 3.2. The experimental setup is described in detail in C.

### D.4 RESULTS FOR MORE MODELS IN S2

In this subsection, we expand the range of models and further validated our conclusions. Tab. 15 presents some experimental results obtained using LLaMA3-8B and LLaMA2-13B (Touvron et al., 2023) on BOSS in S2. We employed the GPTQ method (Frantar et al., 2022) for quantization, keeping the experimental settings consistent with those in Sec. 3.1. We can still observe that the I,I,D results highlighted with background colors have a low overlap with the bolded optimal performance results. This indicates that the same conclusion can be drawn across different series and scales of LLaMA models: the similarity in distribution between calibration data and test datasets does not significantly improve performance.

### D.5 RESULTS FOR FULL PRECISION ON BOSS IN S2

In this subsection, we present the full-precision results for S2 in Tab. 16. As existing quantization methods can achieve performance comparable to full precision in 4-bit setting, we include all full-precision results in the appendix to save space. These full-precision results serve as a baseline for evaluating the impact of quantization on model performance and provide a reference for future research. Overall, although advancements in quantization techniques enable 4-bit models to approach full-precision performance on certain tasks, performance degradation remains a challenge in 3-bit quantization settings.

### D.6 RESULTS FOR 2-BIT ON BOSS IN S2

In this subsection, we present part of the results for the 2-bit quantization in Tab. 17 on BOSS in S2. We observe that at 3 bits, the zero-shot performance experiences a significant decline, while

Table 13: Cross-dataset distribution shift in Chinese domain specific task. To save space, abbreviations are used for datasets. Each row presents the 0-shot and 5-shot experimental results using different datasets as calibration sets on the same test dataset. Results with colored backgrounds indicate I.I.D results, while those without color represent OOD results. The higher the metric, the better the performance. Bold results indicate the best performance on the same test dataset.

| Method | Test | W/A | 0-shot | | 5-shot | | Test | W/A | 0-shot | | 5-shot | |
|---|---|---|---|---|---|---|---|---|---|---|---|---|
| | | | **Calib.** | | | | | | **Calib.** | | | |
| | | | CE-HM | CM-HM | CE-HM | CM-HM | | | CE-HM | CM-HM | CE-HM | CM-HM |
| GPTQ | CE-HM | 4/16 | **39.4** | 37.9 | **53.2** | 52.1 | CM-HM | 4/16 | 50.0 | **50.7** | 59.1 | 59.1 |
| | | 3/16 | **30.0** | 28.0 | 38.1 | **41.9** | | 3/16 | **32.3** | 30.6 | 52.4 | **54.4** |
| | | 2/16 | **25.1** | 24.4 | **23.9** | 23.4 | | 2/16 | **25.3** | 23.7 | **25.9** | 24.4 |
| | Test | W/A | CE-SS | CM-SS | CE-SS | CM-SS | Test | W/A | CE-SS | CM-SS | CE-SS | CM-SS |
| | CE-SS | 4/16 | **36.9** | 35.4 | **58.8** | 57.5 | CM-SS | 4/16 | 53.9 | **54.0** | 63.1 | **63.8** |
| | | 3/16 | **34.6** | 30.3 | **51.9** | 47.5 | | 3/16 | 32.8 | **34.3** | **55.4** | 54.6 |
| | | 2/16 | **25.1** | 23.9 | **25.9** | 24.7 | | 2/16 | 25.7 | **26.2** | **25.6** | 25.3 |
| | Test | W/A | CE-ST | CM-ST | CE-ST | CM-ST | Test | W/A | CE-ST | CM-ST | CE-ST | CM-ST |
| | CE-ST | 4/16 | **30.4** | 26.0 | **41.8** | 39.2 | CM-ST | 4/16 | **39.3** | 35.2 | 43.1 | **43.8** |
| | | 3/16 | **28.1** | 25.7 | 33.9 | **35.5** | | 3/16 | **29.9** | 25.7 | **38.6** | 37.7 |
| | | 2/16 | 24.6 | **25.4** | 24.5 | **25.0** | | 2/16 | **26.2** | 25.7 | 24.5 | **25.2** |
| | Test | W/A | CE-HM | CM-HM | CE-HM | CM-HM | Test | W/A | CE-HM | CM-HM | CE-HM | CM-HM |
| SpQR | CE-HM | 4/16 | **38.5** | 36.3 | **53.8** | 52.5 | CM-HM | 4/16 | 52.9 | 49.3 | 59.0 | **59.5** |
| | | 3/16 | **36.0** | 34.6 | **47.9** | 46.6 | | 3/16 | **49.5** | 38.1 | **57.1** | 56.9 |
| | | 2/16 | 30.1 | **30.9** | **37.4** | 34.5 | | 2/16 | **39.3** | 26.0 | **47.5** | 46.3 |
| | Test | W/A | CE-SS | CM-SS | CE-SS | CM-SS | Test | W/A | CE-SS | CM-SS | CE-SS | CM-SS |
| | CE-SS | 4/16 | 38.2 | **38.9** | **60.0** | 57.7 | CM-SS | 4/16 | 54.8 | 54.3 | 63.8 | **64.7** |
| | | 3/16 | **39.8** | 34.7 | **56.1** | 53.3 | | 3/16 | 52.8 | 51.1 | 59.4 | **60.2** |
| | | 2/16 | 30.1 | **32.1** | **39.5** | 37.3 | | 2/16 | 38.8 | **39.7** | 44.2 | **47.1** |
| | Test | W/A | CE-ST | CM-ST | CE-ST | CM-ST | Test | W/A | CE-ST | CM-ST | CE-ST | CM-ST |
| | CE-ST | 4/16 | **32.2** | 30.3 | **41.5** | 41.1 | CM-ST | 4/16 | **40.4** | 39.5 | **43.7** | 43.3 |
| | | 3/16 | **31.1** | 28.4 | 37.5 | **37.8** | | 3/16 | 37.4 | **37.8** | 40.8 | **41.4** |
| | | 2/16 | **27.8** | 27.7 | **32.2** | 30.6 | | 2/16 | 31.8 | **31.9** | **35.9** | 35.6 |
| | Test | W/A | CE-HM | CM-HM | CE-HM | CM-HM | Test | W/A | CE-HM | CM-HM | CE-HM | CM-HM |
| AWQ | CE-HM | 4/16 | **36.5** | 35.6 | 47.7 | **49.0** | CM-HM | 4/16 | 47.8 | **53.2** | **58.5** | 58.2 |
| | | 3/16 | 26.7 | **29.7** | **41.1** | 40.8 | | 3/16 | 42.6 | **50.5** | 48.0 | **49.5** |
| | | 2/16 | 24.2 | **24.3** | **24.0** | 23.3 | | 2/16 | 25.9 | 42.4 | **25.8** | 23.4 |
| | Test | W/A | CE-SS | CM-SS | CE-SS | CM-SS | Test | W/A | CE-SS | CM-SS | CE-SS | CM-SS |
| | CE-SS | 4/16 | 32.2 | **34.9** | **57.5** | 56.7 | CM-SS | 4/16 | 51.3 | **52.4** | **62.2** | 61.4 |
| | | 3/16 | **32.6** | 31.5 | **42.7** | 40.5 | | 3/16 | 40.1 | **42.1** | 50.5 | **50.8** |
| | | 2/16 | 24.8 | **25.0** | 24.9 | **25.7** | | 2/16 | 24.8 | **24.9** | **24.8** | 24.7 |
| | Test | W/A | CE-ST | CM-ST | CE-ST | CM-ST | Test | W/A | CE-ST | CM-ST | CE-ST | CM-ST |
| | CE-ST | 4/16 | 26.6 | **29.4** | **39.1** | 38.6 | CM-ST | 4/16 | **36.7** | 35.3 | 41.0 | **41.6** |
| | | 3/16 | 26.2 | **27.1** | 31.9 | **34.0** | | 3/16 | **31.7** | **31.7** | **36.3** | 35.5 |
| | | 2/16 | **25.1** | 24.9 | **25.7** | 25.2 | | 2/16 | **24.6** | **24.6** | 24.1 | **24.5** |
| | Test | W/A | CE-HM | CM-HM | CE-HM | CM-HM | Test | W/A | CE-HM | CM-HM | CE-HM | CM-HM |
| SQ | CE-HM | 4/8 | **27.2** | **27.2** | **24.7** | 24.5 | CM-HM | 4/8 | **31.6** | 29.8 | **29.4** | 27.1 |
| | | 3/8 | **25.5** | **25.5** | **24.9** | 23.9 | | 3/8 | 24.7 | **24.8** | **25.3** | 23.9 |
| | | 2/8 | **27.1** | 24.2 | **25.5** | 24.2 | | 2/8 | 24.1 | **25.5** | 24.8 | **25.3** |
| | Test | W/A | CE-SS | CM-SS | CE-SS | CM-SS | Test | W/A | CE-SS | CM-SS | CE-SS | CM-SS |
| | CE-SS | 4/8 | **27.4** | 26.7 | 24.4 | **24.5** | CM-SS | 4/8 | **33.1** | 28.2 | **28.7** | 25.8 |
| | | 3/8 | **26.1** | 25.0 | **26.2** | 24.4 | | 3/8 | 25.0 | **25.1** | **24.7** | 24.6 |
| | | 2/8 | **26.6** | 25.1 | **25.3** | 23.3 | | 2/8 | 24.3 | **25.3** | 25.2 | **25.3** |
| | Test | W/A | CE-ST | CM-ST | CE-ST | CM-ST | Test | W/A | CE-ST | CM-ST | CE-ST | CM-ST |
| | CE-ST | 4/8 | **32.2** | 26.2 | **25.5** | 23.9 | CM-ST | 4/8 | **28.2** | 27.7 | 26.9 | **43.3** |
| | | 3/8 | **31.1** | 27.4 | 24.8 | **25.6** | | 3/8 | **25.4** | 24.2 | 24.4 | **41.4** |
| | | 2/8 | **27.8** | 26.8 | 24.9 | **26.8** | | 2/8 | 24.8 | **24.9** | 24.6 | **35.6** |

few-shot learning can substantially improve the performance of the quantized model. However, at 2 bits, both zero-shot and few-shot performance face a marked deterioration, with few-shot learning no longer able to significantly enhance model performance. We believe it is challenging to derive useful performance insights from results that are nearly zero; therefore, we do not include cases with 2-bit or lower quantization in the distribution shift experiments.

Table 14: Cross-subject distribution shift in Chinese domain-specific task. To save space, abbreviations are used for datasets. Each row presents the experimental results using different datasets as calibration sets on the same test dataset. Results with colored backgrounds indicate I.I.D results, while those without color represent OOD results. The higher the metric, the better the performance. Bold results indicate the best performance on the same test set.

| Meth. | Test | Gene. | W/A | HM | SS | ST | Test | Gene. | W/A | HM | SS | ST | Test | Gene. | W/A | HM | SS | ST |
|---|---|---|---|---|---|---|---|---|---|---|---|---|---|---|---|---|---|---|
| GPTQ | HM | 0-shot | 4/16 | **39.4** | 36.4 | 37.6 | SS | 0-shot | 4/16 | 38.8 | 36.9 | **38.9** | ST | 0-shot | 4/16 | **30.4** | 28.4 | **30.4** |
| | | | 3/16 | 30.0 | **30.5** | 29.2 | | | 3/16 | 29.6 | **34.6** | 30.4 | | | 3/16 | 25.9 | **28.3** | 28.1 |
| | | | 2/16 | 25.1 | 24.1 | **26.2** | | | 2/16 | **27.3** | 25.1 | 25.2 | | | 2/16 | **24.9** | 24.8 | 24.6 |
| | | 5-shot | 4/16 | **53.2** | 52.9 | 52.2 | | 5-shot | 4/16 | **58.9** | 58.8 | 60.1 | | 5-shot | 4/16 | 40.9 | 40.4 | **41.8** |
| | | | 3/16 | 38.1 | **43.5** | 39.9 | | | 3/16 | 42.5 | **51.9** | 48.2 | | | 3/16 | 29.7 | **34.1** | 33.9 |
| | | | 2/16 | 23.9 | **26.2** | 23.7 | | | 2/16 | 24.3 | **25.9** | 24.6 | | | 2/16 | **27.3** | 25.1 | 24.5 |
| SpQR | HM | 0-shot | 4/16 | 38.5 | 38.0 | **40.9** | SS | 0-shot | 4/16 | 39.3 | 38.2 | **41.3** | ST | 0-shot | 4/16 | 30.3 | 29.9 | **32.2** |
| | | | 3/16 | 36.0 | **39.0** | 38.9 | | | 3/16 | 34.8 | **39.8** | 39.0 | | | 3/16 | 30.5 | 29.1 | **31.1** |
| | | | 2/16 | **30.1** | 29.9 | 29.2 | | | 2/16 | 28.7 | 30.1 | **30.6** | | | 2/16 | 26.1 | 26.6 | **27.8** |
| | | 5-shot | 4/16 | **53.8** | 51.0 | 52.6 | | 5-shot | 4/16 | 59.3 | **60.0** | 59.6 | | 5-shot | 4/16 | 41.4 | 41.0 | **41.5** |
| | | | 3/16 | **47.9** | 45.8 | 46.5 | | | 3/16 | 52.8 | **56.1** | 53.0 | | | 3/16 | 36.5 | **37.6** | 37.5 |
| | | | 2/16 | 37.4 | 35.0 | **37.7** | | | 2/16 | 40.6 | 39.5 | **45.0** | | | 2/16 | 28.3 | 28.0 | **32.2** |
| AWQ | HM | 0-shot | 4/16 | **36.5** | 34.2 | 33.4 | SS | 0-shot | 4/16 | **35.2** | 32.2 | 31.4 | ST | 0-shot | 4/16 | **28.5** | **28.5** | 26.6 |
| | | | 3/16 | 26.7 | **32.1** | 27.5 | | | 3/16 | 28.3 | **32.6** | 28.2 | | | 3/16 | 27.7 | **28.9** | 26.2 |
| | | | 2/16 | 24.2 | 24.2 | **24.6** | | | 2/16 | 24.9 | 24.8 | **25.2** | | | 2/16 | 24.9 | 24.8 | **25.1** |
| | | 5-shot | 4/16 | 47.7 | 49.7 | **51.2** | | 5-shot | 4/16 | 53.4 | **57.5** | 56.6 | | 5-shot | 4/16 | 37.7 | 38.5 | **39.1** |
| | | | 3/16 | **41.1** | 38.4 | 37.4 | | | 3/16 | **44.0** | 42.7 | 38.7 | | | 3/16 | **31.9** | 31.0 | **31.9** |
| | | | 2/16 | 24.0 | **24.6** | 23.8 | | | 2/16 | 23.9 | 24.9 | **25.1** | | | 2/16 | 25.2 | 25.3 | **25.7** |
| SQ | HM | 0-shot | 4/8 | 27.2 | **28.9** | 27.4 | SS | 0-shot | 4/8 | **28.3** | 27.4 | 28.2 | ST | 0-shot | 4/8 | 26.8 | **28.0** | 25.4 |
| | | | 3/8 | 25.5 | 23.9 | **26.4** | | | 3/8 | **26.4** | 26.1 | 25.5 | | | 3/8 | 26.6 | 25.2 | **26.7** |
| | | | 2/8 | **27.1** | 25.2 | 24.8 | | | 2/8 | 26.2 | **26.6** | 26.4 | | | 2/8 | **26.4** | **26.4** | 25.7 |
| | | 5-shot | 4/8 | 24.7 | 24.2 | **24.9** | | 5-shot | 4/8 | **26.0** | 24.4 | 24.3 | | 5-shot | 4/8 | 24.8 | 24.3 | **25.5** |
| | | | 3/8 | 24.9 | **26.4** | 26.2 | | | 3/8 | 24.7 | **26.2** | 25.9 | | | 3/8 | **26.8** | 25.3 | 24.8 |
| | | | 2/8 | 25.5 | **26.4** | 24.2 | | | 2/8 | **26.5** | 25.3 | 24.9 | | | 2/8 | 26.6 | **26.8** | 24.9 |

Table 15: Cross-dataset distribution shift evaluation on BOSS in S2. The models used for these experiments are LLaMA3-8B and LLaMA2-13B. "Calib." represents the calibration dataset, and "Gene." represents generalization scenario. To save space, abbreviations are used for datasets. Each row presents experimental results using different datasets as calibration sets on the same test dataset. Results with colored backgrounds indicate I.I.D results, while those without color represent OOD results. The higher the metric, the better the performance. Bold results indicate the best performance on the same test dataset.

| Model | Test | Gene. | W/A | AZ | DS | SE | SST | Model | Test | Gene. | W/A | AZ | DS | SE | SST |
|---|---|---|---|---|---|---|---|---|---|---|---|---|---|---|---|
| LLaMA3-8B | AZ | 0-shot | 4/16 | 85.15 | **88.06** | 81.24 | 86.77 | LLaMA2-13B | AZ | 0-shot | 4/16 | **88.74** | 79.73 | 87.59 | 88.69 |
| | | | 3/16 | 0.00 | 0.00 | 0.00 | 0.00 | | | | 3/16 | 0.01 | 81.49 | 81.64 | **85.76** |
| | | 3-shot | 4/16 | 85.67 | 85.73 | 85.74 | **85.81** | | | 3-shot | 4/16 | 81.64 | 81.05 | **84.44** | 82.05 |
| | | | 3/16 | 0.00 | 0.00 | 0.00 | 0.00 | | | | 3/16 | 44.74 | **86.57** | 86.07 | 85.71 |
| | DS | 0-shot | 4/16 | 56.94 | 58.76 | 54.88 | **59.03** | | DS | 0-shot | 4/16 | **62.79** | 40.52 | 57.99 | 62.76 |
| | | | 3/16 | 0.00 | 0.00 | **1.57** | 0.00 | | | | 3/16 | 0.00 | **61.04** | 48.03 | 53.73 |
| | | 3-shot | 4/16 | 54.78 | 56.84 | 56.09 | **60.55** | | | 3-shot | 4/16 | 47.14 | 24.35 | **53.46** | 48.51 |
| | | | 3/16 | 0.00 | 0.00 | 0.00 | 1.34 | | | | 3/16 | 49.73 | 64.38 | 63.43 | **65.25** |
| | SE | 0-shot | 4/16 | 43.35 | **45.80** | 37.08 | 45.50 | | SE | 0-shot | 4/16 | 42.29 | 21.30 | 37.27 | **44.55** |
| | | | 3/16 | 0.00 | 0.00 | **0.32** | 0.00 | | | | 3/16 | 0.00 | 38.03 | 36.01 | **44.81** |
| | | 3-shot | 4/16 | 51.15 | 49.74 | 45.19 | **51.22** | | | 3-shot | 4/16 | 54.27 | 42.60 | 50.56 | **55.44** |
| | | | 3/16 | 0.00 | 0.00 | 0.00 | 0.00 | | | | 3/16 | 34.20 | 52.04 | **56.30** | 48.11 |
| | SST | 0-shot | 4/16 | 61.02 | **62.84** | 55.28 | 61.41 | | SST | 0-shot | 4/16 | 62.32 | 18.64 | 43.55 | **64.41** |
| | | | 3/16 | 0.00 | 0.00 | 1.56 | 0.00 | | | | 3/16 | 0.00 | **64.02** | 50.33 | 63.62 |
| | | 3-shot | 4/16 | 61.54 | 60.37 | 61.80 | **67.67** | | | 3-shot | 4/16 | 32.07 | 17.47 | **33.51** | 31.03 |
| | | | 3/16 | 0.00 | 0.00 | 0.00 | **0.26** | | | | 3/16 | 26.86 | **64.15** | 59.19 | 53.85 |

Table 16: Full precision results on BOSS in S2. "Gene." represents generalization scenario. To save space, abbreviations are used for datasets.

| EQA | | | | | | SA | | | | | |
|---|---|---|---|---|---|---|---|---|---|---|---|
| **Gene.** | **W/A** | **SQ** | **AQA** | **NQA** | **SQA** | **Gene.** | **W/A** | **AZ** | **DS** | **SE** | **SST** |
| **0-shot** | 16/16 | 54.00 | 28.68 | 39.54 | 46.34 | **0-shot** | 16/16 | 74.75 | 50.40 | 27.64 | 45.76 |
| **Few-shot** | 16/16 | 67.93 | 37.21 | 49.40 | 62.41 | **Few-shot** | 16/16 | 79.90 | 53.13 | 43.72 | 55.54 |
| **NLI** | | | | | | **TD** | | | | | |
| **Gene.** | **W/A** | **MN** | **AN** | **MN** | **CN** | **Gene.** | **W/A** | **CC** | **AC** | **IH** | **TG** |
| **0-shot** | 16/16 | 0.47 | 1.55 | 0.14 | 0.06 | **0-shot** | 16/16 | 62.25 | 17.60 | 48.94 | 59.69 |
| **Few-shot** | 16/16 | 44.81 | 33.72 | 43.32 | 34.62 | **Few-shot** | 16/16 | 91.25 | 16.44 | 63.76 | 73.59 |

Table 17: 2-bit results on BOSS in S2. We present some results of Sentiment Analysis. "Gene." represents generalization scenario. To save space, abbreviations are used for datasets.

| Calib. | Gene. | SQUAD | AdvQA | NeswQA | SearchQA |
|---|---|---|---|---|---|
| **AdvQA** | 0-shot | 0.46 | 0.22 | 0.39 | 0.07 |
| | 1-shot | 0.52 | 0.22 | 0.20 | 0.06 |
| **SearchQA** | 0-shot | 1.35 | 0.76 | 0.70 | 0.90 |
| | 1-shot | 1.86 | 1.02 | 1.18 | 1.20 |

