# OpenReview forum: "Evaluating the Generalization Ability of Quantized LLMs: Benchmark, Analysis, and Toolbox"
_ICLR.cc/2025/Conference — Submitted to ICLR 2025_

### Official Review · Reviewer_QsUE · 2024-10-29

**Soundness:** 2
**Presentation:** 3
**Contribution:** 2
**Rating:** 6
**Confidence:** 3

**Summary:**

This paper addresses the gap in understanding how data impacts the generalization abilities of quantized large language models (LLMs). By benchmarking with over 40 datasets and experimenting with popular LLMs, the study reveals the non-optimal performance of models quantized with calibration data matching the test data distribution. Additionally, the authors provide a modular toolbox to support future exploration into LLM quantization.

**Strengths:**

1. Very large experimental workload. The authors implemented a Python package integrating various LLM quantization models, based on which a large number of experimental results were measured.
2. From the perspectives of IID and OOD, detailed experimental data provide useful insight into the generalization performance of quantifying LLM.

**Weaknesses:**

1. This paper does not propose new algorithms but rather tests quantization algorithms proposed by other researchers before. Can the authors provide some more insights, such as: does the generalization performance of different quantization algorithms differ?
2. Eq1 simply uses the number of samples where the performance of the I.I.D calibration set exceeds that of the OOD to evaluate, which is actually a little crude. LLM evaluation is a dirty task, and the accuracy of only higher a little does not mean that the model is better. This will weaken the validity of the paper's conclusions. It is recommended to have some statistical technical hypotheses and tests (like the Box-and-Whisker Plot or standard deviation).

**Questions:**

1. Will MI continue to be developed to support new LLM quantization algorithms?
2. L462-L464, the authors utilize a dataset consisting of 128 random segments and each containing 512 tokens. This is actually a bit odd, as 128*2048 token length calibration sets are more common. Therefore, does the size of the calibration set affect the generalization performance of the quantization model? For example, different sequence numbers (e.g. 1,16, 128, 512, 1024) and lengths (e.g. 128, 512, 1024, 2048).
3. Similarly, the author mainly discusses the 7B-13B size model in this paper. Will the conclusion change for the 70B+ model? Intuitively, the 70B model would be more redundant and easier to quantify.

---

> ### Author Response · Authors · 2024-11-13
>
> > W1: This paper does not propose new algorithms but rather tests quantization algorithms proposed by other researchers before. Can the authors provide some more insights, such as: does the generalization performance of different quantization algorithms differ?
>
> Thank you for your suggestion!
>
> This paper primarily focuses on evaluating the effects of distribution shifts from calibration to test sets on quantized LLMs, rather than on algorithm design. In future work, we will build on this research to develop algorithms that improve calibration set selection, optimizing quantized model performance from a data perspective—an aspect that has not been explored in the quantization field thus far.
>
> Due to space constraints, we did not elaborate on additional findings in the paper, so I’ll provide some supplementary insights here.
>
> For experiments in S1 (Figure 2), we observe that GPTQ experiences significant performance drops at lower bits, such as 2-bit, whereas SPQR does not exhibit this issue and may even show performance gains. SPQR is specifically designed for low-bit scenarios, which may be related to its capability to identify and isolate outlier weights.
>
> In the cross-dataset distribution shift experiments in S2 on BOSS (Table 2), we find that at 4-bit, GPTQ, SPQR, and AWQ achieve performance close to full precision, but at 3-bit, both GPTQ and AWQ suffer notable performance losses. Additionally, SmoothQuant consistently shows a greater performance drop relative to full precision, likely due to activation quantization, indicating that activation quantization remains a challenging issue. It is also noteworthy that while 3-bit shows minimal performance drop in S1, it encounters significant losses on BOSS, suggesting that quantized LLMs generally exhibit lower generalization on the BOSS dataset compared to datasets in S1.
> From the prompt paradigm perspective, few-shot settings significantly recover model performance, restoring accuracy by up to 40%, demonstrating the performance-boosting effect of in-context learning for LLMs.
> Regarding distribution shift characteristics, we observe that calibration dataset suitability varies across algorithms. For example, with GPTQ, the SQA dataset performs better on SQ tasks, while no single dataset stands out for AWQ. This variation may stem from differences in how each algorithm leverages calibration data internally.
>
> Overall, from an algorithmic perspective, **generalization capability varies significantly across quantization algorithms**.

---

> ### Author Response · Authors · 2024-11-23
>
> > Q1：Will MI continue to be developed to support new LLM quantization algorithms?
>
> Thank you for your question. Yes, we plan to continue the development of MI-optimize to support new LLM quantization algorithms, models and datasets, including methods such as pruning and distillation. As quantization techniques and LLMs evolve, MI-optimize will be updated to accommodate new quantization methods while ensuring effective interpretability of compressed models. Additionally, MI-optimize will be leveraged in the development and performance evaluation of new quantization algorithms, helping researchers understand the impact of quantization on model behavior.
>
> Thank you for your attention, and we look forward to continuing to improve MI-optimize in our future work.

---

> ### Author Response · Authors · 2024-11-25
>
> > Q2: L462-L464, the authors utilize a dataset consisting of 128 random segments and each containing 512 tokens. This is actually a bit odd, as 128*2048 token length calibration sets are more common. Therefore, does the size of the calibration set affect the generalization performance of the quantization model? For example, different sequence numbers (e.g. 1,16, 128, 512, 1024) and lengths (e.g. 128, 512, 1024, 2048).
>
> Thank you for your question!
>
> First, we would like to clarify that the experiments conducted in S1 and S2 were performed using a token length of 128\*2048. In contrast, the algorithm fusion experiments in Section 4 were carried out using a token length of 128\*512 to test the new features of our toolbox. It's important to note that when using downstream task datasets as calibration sets, the samples consist of questions with varying lengths. Specifically, we use a truncation length of 2048 tokens, rather than a fixed 2048-token length.
>
> Regarding the size of the calibration set, we follow a widely accepted standard: using 128 samples as the calibration set. For algorithms with slightly lower performance, increasing the calibration set size can enhance the model's performance. However, this also faces diminishing returns, meaning that simply increasing the size does not consistently improve performance. For example, when evaluating perplexity (PPL) on datasets like C4, 16 samples represent the performance inflection point [1]. For algorithms like GPTQ, increasing the calibration set size does not result in a significant performance boost [1].
>
> Additionally, we have evaluated the GPTQ algorithm on our own benchmark. The table below shows the performance changes as the sample size varies. From the data, we can observe that increasing the number of calibration samples does not lead to a noticeable improvement in performance.
>
> | Dataset\Samples |1|16|64|128|256|512|
> | --------------- | ---- | --- | --- | -------- | --- | -------- |
> | SA-DS |51.59|47.56|50.17|45.02|51.54|50.40|
> |EQA-AQ|33.91|35.66|34.95|32.77|36.30|35.23|
>
> [1] On the Impact of Calibration Data in Post-training Quantization and Pruning.

---

> ### Author Response · Authors · 2024-11-25
>
> > Q3: Similarly, the author mainly discusses the 7B-13B size model in this paper. Will the conclusion change for the 70B+ model? Intuitively, the 70B model would be more redundant and easier to quantify.
>
> Thank you for your valuable feedback!
>
> The issue you raised regarding 70B+ models is indeed very insightful. In this paper, we focused primarily on models in the 7B-13B range to demonstrate key concepts that are generally applicable across different model sizes. However, we also recognize that as the model size increases, different behaviors and dynamics may emerge. We plan to further explore the performance of larger models in future work and will include experiments with the 70B model in the final version of the paper.

---

> ### Author Response · Authors · 2024-11-25
>
> > W2: Eq1 simply uses the number of samples where the performance of the I.I.D calibration set exceeds that of the OOD to evaluate, which is actually a little crude. LLM evaluation is a dirty task, and the accuracy of only higher a little does not mean that the model is better. This will weaken the validity of the paper's conclusions. It is recommended to have some statistical technical hypotheses and tests (like the Box-and-Whisker Plot or standard deviation).
>
> We sincerely appreciate your valuable comment!
>
> We acknowledge that the current method may have limitations in assessing the performance differences between IID and OOD settings, particularly when the performance gap is small. To address this, we have visualized the results using box-and-whisker plots, which show that the performance distributions are similar, supporting our original conclusions. To further ensure the validity of our experiments, we plan to conduct repeated evaluations and compute variance and confidence intervals in future work to improve the robustness of the results.

---

> > ### Author Response · Authors · 2024-11-26
> >
> > In addition, we conducted a **Wilcoxon Signed-Rank Test** on the data in **Table 2** and examined whether there were significant differences between the paired samples of the I.I.D and OOD datasets.
> >
> > Null hypothesis (H₀): There is no significant difference in performance between the I.I.D and OOD settings.
> >
> > Alternative hypothesis (H₁): There is a significant difference in performance between the I.I.D and OOD settings.
> >
> > By setting the significance level ( α = 0.05), we calculated the p-values for each method and each task, as shown in the table below:
> >
> > | Algorithm\Dataset |  EQA  |  SA   |  NLI  |  TD   |
> > |:-----------------:|:-----:|:-----:|:-----:|:-----:|
> > |       GPTQ        | 0.433 | 0.252 | 0.155 | 0.821 |
> > |       SpQR        | 0.860 | 0.940 | 0.016 | 0.348 |
> > |        AWQ        | 0.850 | 0.850 | 0.594 | 0.669 |
> > |    Smoothquant    | 0.075 | 0.612 | 0.893 | 0.286 |
> >
> > We can observe that all the p-values are nearly greater than the significance level α, indicating that we fail to reject the null hypothesis (H₀), and **there is no significant difference in performance between the I.I.D and OOD settings**.

---

> ### Author Response · Authors · 2024-11-25
>
> As a kind reminder, the discussion period is drawing close. Please let us know if there remains anything that we can further clarify to improve our work. Many thanks in advance.

---

### Official Review · Reviewer_U3h9 · 2024-11-03

**Soundness:** 3
**Presentation:** 3
**Contribution:** 2
**Rating:** 5
**Confidence:** 4

**Summary:**

This paper explores the generalization performance of quantized LLMs and introduces a comprehensive benchmark suite alongside a modular toolbox. The study examines how calibration data distribution affects generalization, revealing two key insights:
1. Tasks exhibit varying sensitivity to quantization, with some tasks showing improved performance under low-bit quantization.
2. Consistency between calibration and test data distributions does not consistently yield optimal performance.

**Strengths:**

- **Extensive Empirical Evaluation**: The study conducts comprehensive experiments across multiple datasets and quantization methods, providing valuable insights into LLM generalization under different calibration scenarios.
- **Practical Contribution**: The proposed modular toolbox is a significant resource for the evaluation and application of quantized LLMs, potentially benefiting the broader research community.

**Weaknesses:**

1. **Lack of Guidance on Calibration Data Selection**: Although the paper presents intriguing findings, it does not offer concrete criteria or methods for selecting calibration data to enhance the generalization of quantized LLMs. This limits its practical impact and novelty.
2. **Visualization Issues**:
    - Radar charts (Figures 2, 5, and 6) lack marked magnitudes for the scores on the radius, and text overlays reduce clarity.
    - The task types, while indicated by background colors, are not explicitly labeled. An additional legend would make the visualizations more intuitive.

**Questions:**

1. The paper highlights task-specific sensitivities to quantization. Could the authors provide more detailed analysis or theoretical insights into why some tasks are more robust than others?
2. Given that the evaluation could potentially be performed by extending existing toolboxes, what is the necessity of developing a new quantization and evaluation framework?

---

> ### Author Response · Authors · 2024-11-13
>
> > Q2: Given that the evaluation could potentially be performed by extending existing toolboxes, what is the necessity of developing a new quantization and evaluation framework?
>
> Thank you for your question!
>
> To the best of our knowledge, no existing toolbox provides a framework for evaluating distributional shifts from calibration to test sets to assess the generalization ability of quantized models. Additionally, our toolbox implements a wider range of algorithms and datasets, enabling more extensive testing of quantized models. Our toolbox also supports the combination of multiple quantization algorithms, achieving better performance than using any single algorithm alone.

---

> > ### Comment · Reviewer_U3h9 · 2024-11-26
> >
> > Thank you for providing further details about your toolbox. While I recognize the value of creating a unified and modular framework, I remain unconvinced that the specific task of evaluating distributional shifts between calibration and test sets requires a dedicated "framework". This task can be achieved through preprocessing datasets, adapting configurations, and utilizing existing tools for quantization and evaluation. Given that many quantization algorithms are already open-sourced, integrating these with preprocessing pipelines may suffice for such evaluations without the need for a standalone toolbox.
> >
> > The claim that the toolbox supports combining multiple quantization algorithms is intriguing. I thought many quantization methods are typically exclusive in how they modify model weights and activations, making simultaneous application challenging. The practical implementation of such combinations raises questions about compatibility and performance benefits. Could the author provide concrete examples or case studies where combining quantization algorithms yields improved performance?
> >
> > The broader algorithm and dataset support within your toolbox is a notable engineering contribution, enabling extensive evaluations. However, the academic novelty and contribution remain limited. Considering these limitations, I will maintain my current score.

---

> > > ### Author Response · Authors · 2024-11-30
> > >
> > > Thank you very much for your detailed response once again!
> > >
> > > Our toolbox allows for the combination of different algorithms, such as: 1) using different algorithms within the same layer, e.g., using SmoothQuant for quantizing activations and GPTQ for quantizing weights; 2) using different quantization algorithms across different layers, e.g., some layers use GPTQ while others use AWQ. The innovation lies in the fact that other frameworks do not support such granular combinations. The advantage of our toolbox is that, within the same layer, different quantization algorithms can be combined to leverage the strengths of each algorithm. For instance, SmoothQuant has the advantage of smoothing outliers when quantizing activations, while GPTQ excels at quantizing weights. Different layers may have different sensitivities to quantization algorithms, and using the most suitable algorithm for each layer can help minimize quantization loss.
> > >
> > > The table below demonstrates the performance differences between using SmoothQuant for both activations and weights, and using SmoothQuant for activations and GPTQ for weights. We can observe that switching to GPTQ for quantizing weights improves performance, which suggests that in the future, we can reasonably adopt a combination of quantization methods to optimize the strengths and mitigate the weaknesses of each.
> > >
> > > |       Method\PPL(↓)       | Wiki2 |   C4    |  PTB  |
> > > |:----------------------:|:-----:|:-------:|:-----:|
> > > |    Smoothquant(W+A)    | 34.87 | 5133.82 | 20.82 |
> > > | Smoothquant(A)+GPTQ(W) | 22.95 | 1359.59 | 13.39 |
> > >
> > > In the future, we will continue to conduct more detailed analyses of the algorithmic combinations, including using different algorithms based on the characteristics of different layers, applying different methods for weights and activations, and so on. Additionally, we will integrate more advanced and updated algorithms to enhance our toolbox.
> > >
> > > If we have successfully addressed any of your concerns, may we kindly ask you to reconsider the score and potentially raise it? We sincerely appreciate your thoughtful feedback and continued support, and we look forward to any further suggestions or comments you may have.

---

> ### Author Response · Authors · 2024-11-13
>
> > W2: Visualization Issues:
> Radar charts (Figures 2, 5, and 6) lack marked magnitudes for the scores on the radius, and text overlays reduce clarity.
> The task types, while indicated by background colors, are not explicitly labeled. An additional legend would make the visualizations more intuitive.
>
> Thank you very much for your valuable suggestion!
>
> In future versions, we will optimize these images to enhance clarity and readability, allowing readers to more easily see specific values and task types.

---

> ### Author Response · Authors · 2024-11-23
>
> > W1: Lack of Guidance on Calibration Data Selection: Although the paper presents intriguing findings, it does not offer concrete criteria or methods for selecting calibration data to enhance the generalization of quantized LLMs. This limits its practical impact and novelty.
>
> Thank you for your comments!
>
> Our recommendation is to use a high-quality corpus as the calibration set to recover the performance loss of large models due to quantization. We discuss this in detail in **Section 3.3** and provide references for future work. In **Table 4**, we show the performance differences between using a high-quality corpus and using a downstream task dataset as the calibration set, and we observe that the results are nearly identical. This suggests that using a calibration set with the same distribution as the test set does not significantly improve performance. The large model is relatively robust to the choice of calibration dataset in terms of distribution. Finding the optimal calibration dataset is a direction for further exploration in future work; however, the focus of this paper is not on developing methods for selecting the optimal calibration set, but rather on studying the impact of distribution shifts between the calibration set and the test set on performance.

---

> > ### Comment · Reviewer_U3h9 · 2024-11-26
> >
> > Thank you for your detailed reply. I appreciate the efforts to clarify the focus and contributions of the paper. The focus on distribution shifts between calibration and test sets, appears to address a niche or secondary concern in the broader context of quantized LLMs. The finding that such distribution shifts have minimal impact, while useful to confirm, is not particularly surprising given prior understanding of LLM robustness. This limits the work’s academic contribution, as the main takeaway does not significantly alter existing paradigms or offer novel insights into LLM compression.
> >
> > While the extensive experiments and benchmarks are thorough, they primarily confirm expected outcomes (e.g., LLM robustness to calibration data distribution). The insights into task sensitivity, while interesting, align with established scaling law behaviors and do not push the boundaries of our understanding of quantization.

---

> ### Author Response · Authors · 2024-11-23
>
> > Q1: The paper highlights task-specific sensitivities to quantization. Could the authors provide more detailed analysis or theoretical insights into why some tasks are more robust than others?
>
> The task-specific sensitivity to quantization can be analyzed from the perspective of task complexity. Generally, more complex tasks involve intricate reasoning and decision-making processes, which require scaling laws to address. When LLMs are quantized, the number of parameters decreases, resulting in a smaller model, which leads to a decline in performance. This manifests as a higher sensitivity to quantization for these tasks, with performance degradation being more noticeable—such as in mathematical reasoning tasks.
> On the other hand, simpler tasks can often be handled by smaller models, and even after quantization, the performance remains relatively unaffected. Therefore, quantization has a smaller impact on tasks like natural language inference. Notably, for sentiment analysis tasks, performance improvement is observed because the dataset used is relatively simple, and even small models can achieve good performance.
>
> This aligns with the "**scaling law**" in large models, which indicates that for more complex tasks, increasing model size leads to significant performance improvements. In summary, during the compression process, reducing the number of model parameters has a more pronounced negative effect on complex tasks, resulting in more significant performance degradation.
>
> We hope these explanations help further clarify the significance of our work. Thank you for your valuable feedback.

---

> ### Author Response · Authors · 2024-11-25
>
> As a kind reminder, the discussion period is drawing close. Please let us know if there remains anything that we can further clarify to improve our work. Many thanks in advance.

---

### Official Review · Reviewer_a5T8 · 2024-11-03

**Soundness:** 3
**Presentation:** 3
**Contribution:** 2
**Rating:** 3
**Confidence:** 5

**Summary:**

This paper delves into the impact of the calibration set on the generative capacity of quantized LLMs through extensive experiments. In addition, a novel modular-designed toolbox is proposed to decouple the model quantization pipeline into seperate components to help investigate the different modules.

**Strengths:**

1. This paper thoroughly considers a vast array of datasets and scenarios, which make clear and effective distinctions, to support its experimental conclusions.

2. The quantization methods adopted are all currently mainstream, demonstrating the universality of the experimental discoverages.

**Weaknesses:**

1. A serious issue is that the authors claim that this article is the first to study the impact of the calibration set on the generative capacity of quantized large models. However, to my knowledge, similar work has already been done previously [1]. Therefore, the authors' statement is quite inappropriate.

2. The number of LLMs using for quantizing in the experiment is too small, and their size is relatively small (7B). This limits the generality of the experimental results to a certain extent.

3. This article appears to be a superficial description and summary of experimental phenomena, lacking in-depth discussion.

**Questions:**

1. For W1, apart from revising their statement, the authors also need to provide a detailed description of the differences between their research and the mentioned paper. Since the objectives and main content of this work and the mentioned one are extremely similar, failing to provide clear distinctions is a significant issue.

2. For W2, in the past research, it has been proven that larger LLMs are less sensitive to quantization. Therefore, due to the time constraint of the rebuttal, there is no need to extend the experimental models to various sizes. It suffices to provide experimental data on the largest model (e.g., llama2-70B) to demonstrate that the current conclusions remain valid.

3. For W3, one of the main contributions of a benchmark is to provide guidance for future work. Therefore, the authors should offer some appropriate suggestions based on the experimental results. For example, they should recommend which dataset is best suited as a calibration set for future quantization methods to achieve optimal results. In more detail, although different calibration sets may yield varying results, a comprehensively optimal dataset should be selected for calibration.

4. The athors should further describe the definition of IID and OOD which appears abruptly. Does IID means the different or the same dataset under the same subject?


[1] Miles Williams and Nikolaos Aletras. 2024. On the Impact of Calibration Data in Post-training Quantization and Pruning. In Proceedings of the 62nd Annual Meeting of the Association for Computational Linguistics (Volume 1: Long Papers), pages 10100–10118, Bangkok, Thailand. Association for Computational Linguistics.

---

> ### Author Response · Authors · 2024-11-13
>
> > Q1: For W1, apart from revising their statement, the authors also need to provide a detailed description of the differences between their research and the mentioned paper. Since the objectives and main content of this work and the mentioned one are extremely similar, failing to provide clear distinctions is a significant issue.
>
> Thank you for your comment!
>
> First, we want to clarify that we **did not claim to be the first to study the impact of calibration datasets**. Could you please indicate the specific location where this may have been misunderstood? Our work is the *first to investigate distribution shifts between calibration and test datasets on quantized large LLMs, as well as the first to explore cross-subject distrbution shift experiments*. This fills a gap in assessing the generalization capability of quantized LLMs.
>
> Additionally, we discuss this paper in **lines 490 and 505 in Section 5** and cite it in **line 745**, and **Table 7** highlights the distinctions between our work and prior studies. In particular, [1] primarily uses datasets from the pre-training corpus, employing different samples as calibration datasets for downstream tasks. However, their work does not include evaluations under I.I.D and OOD setting, nor does it extend calibration datasets from pre-training corpus data to downstream task datasets. It also does not address the distribution shift between calibration and test datasets, remaining limited to the S1 setting in our study. In contrast, our work not only varies the use of calibration datasets but also considers how distribution shifts between calibration and test datasets impact quantized LLMs, covering both the S1 and S2 settings outlined in our text.
>
> [1] Miles Williams and Nikolaos Aletras. 2024. On the Impact of Calibration Data in Post-training Quantization and Pruning. In Proceedings of the 62nd Annual Meeting of the Association for Computational Linguistics (Volume 1: Long Papers), pages 10100–10118, Bangkok, Thailand. Association for Computational Linguistics.

---

> ### Author Response · Authors · 2024-11-13
>
> > Q3: For W3, one of the main contributions of a benchmark is to provide guidance for future work. Therefore, the authors should offer some appropriate suggestions based on the experimental results. For example, they should recommend which dataset is best suited as a calibration set for future quantization methods to achieve optimal results. In more detail, although different calibration sets may yield varying results, a comprehensively optimal dataset should be selected for calibration.
>
> Thanks for this comment.
>
> Our suggestion is that, if you want to obtain a golden dataset, it is better to look for a dataset whose distribution is closer to that of high-quality pretraining corpora, rather than one that is more similar to the test data. This approach can yield better average performance. Using high-quality corpora as the calibration set can recover the loss caused by quantization in large models. We discuss this in detail in **Section 3.3** and provide guidance for future work. In **Table 4**, we show the performance difference between using high-quality corpora and using downstream task datasets as calibration sets. The results are almost identical, suggesting that using a calibration set with the same distribution as the test set does not significantly improve performance. The selection of the calibration dataset for large models is relatively robust in terms of distribution.
>
> For those aiming to obtain an optimal calibration dataset, several factors need to be considered: the quantization algorithm, model, and task. Only by carefully considering these factors can you achieve the best possible calibration dataset and optimize performance. However, this also incurs significant costs. Therefore, there is a trade-off between the performance of the calibration dataset and its associated costs.

---

> ### Author Response · Authors · 2024-11-13
>
> > Q4: The athors should further describe the definition of IID and OOD which appears abruptly. Does IID means the different or the same dataset under the same subject?
>
> Thanks for this suggestion.
>
> I.I.D (Independent and Identically Distributed) refers to a set of random variables that are both independent and follow the same probability distribution, meaning that each data point's value is unaffected by others and all data points come from the same distribution. OOD (Out-of-Distribution) refers to data whose distribution lies outside the range of the distribution seen by the model during training, meaning that the features or distribution of the test data differ from the training data, often leading to poor performance on such data. The concepts of I.I.D and OOD are well-established in the CV field but are relatively underdeveloped in NLP.
>
> In this paper, our definitions of I.I.D. and OOD settings follow the settings in [1] and are extended to consider the distribution shift from the calibration set to the test set. We explain the I.I.D. and OOD settings used in this experiment in **Section 1** and **Figure 1**. The definition of the I.I.D. setting is: samples are drawn using the same sampling strategy from the same sample space. The definition of the sample space is: consisting of data from the same dataset or from the same domain within the same dataset. Data that does not meet this criterion is considered to belong to the OOD setting.
>
> For the cross-dataset distribution shift experiment conducted on BOSS in Section 2, data from the same dataset is considered I.I.D. data. For example, in the EQA task, when using SQ as the calibration set, the test set using SQ is an I.I.D. setting, while the rest are OOD settings.
>
> For the cross-dataset distribution shift experiment on Chinese domain-specific datasets in S2, data from the same dataset is also I.I.D. data. For example, when using C-EVAL as the calibration set, the test set using C-EVAL is an I.I.D. setting, while the rest are OOD settings.
>
> For the cross-subject distribution shift experiment on Chinese domain-specific datasets in S2, data from the same dataset and the same domain is considered I.I.D. data. For example, when using the HM subject from C-EVAL as the calibration set, the test set using the HM subject from C-EVAL is an I.I.D. setting, while all other settings are OOD settings.
>
> [1]LifanYuan,YangyiChen,GanquCui,HongchengGao,FangyuanZou,XingyiCheng,HengJi,ZhiyuanLiu,andMaosongSun.Revisiting out-of-distribution robustness in nlp: Benchmarks,analysis,and llms evaluations. AdvancesinNeuralInformationProcessingSystems,36,2024.

---

> ### Author Response · Authors · 2024-11-13
>
> > W1: A serious issue is that the authors claim that this article is the first to study the impact of the calibration set on the generative capacity of quantized large models. However, to my knowledge, similar work has already been done previously [1]. Therefore, the authors' statement is quite inappropriate.
>
> Thank you for your careful review!
>
> We would like to emphasize that we do not claim to be the first to study the impact of calibration datasets on quantized LLMs. Rather, we are the first to investigate distribution shifts between calibration and test datasets on quantized LLMs, as well as to conduct cross-subject distribution shift experiments, filling a gap in evaluating the generalization ability of quantized LLMs. If there is content suggesting otherwise, please indicate the specific location. We also discuss this paper [1] on lines 490 and 505 in Section 5, and Table 7 highlights the distinctions between our work and all prior studies, as noted in our response to Q1.
>
> [1] Miles Williams and Nikolaos Aletras. 2024. On the Impact of Calibration Data in Post-training Quantization and Pruning. In Proceedings of the 62nd Annual Meeting of the Association for Computational Linguistics (Volume 1: Long Papers), pages 10100–10118, Bangkok, Thailand. Association for Computational Linguistics.

---

> ### Author Response · Authors · 2024-11-13
>
> > W2: The number of LLMs using for quantizing in the experiment is too small, and their size is relatively small (7B). This limits the generality of the experimental results to a certain extent.
>
> Thank you for your comment!
>
> First, we would like to note that we have also tested **LLaMA2-13B** and presented the results in **Table 15**, thereby expanding the range of models. Additionally, we are currently supplementing experiments with larger models to ensure a more comprehensive and robust evaluation.

---

> ### Author Response · Authors · 2024-11-13
>
> > W3: This article appears to be a superficial description and summary of experimental phenomena, lacking in-depth discussion.
>
> Thank you for your suggestion!
>
> In **Section 3.3**, we conducted an in-depth analysis that includes summarizing conclusions, formulating hypotheses, and conducting validations. First, we concluded from the experiments that: Consistency between calibration data and test distribution does not always yield optimal performance. Then, we proposed the hypothesis that LLMs may not require highly relevant data related to downstream tasks to recover performance loss due to quantization. Subsequently, we compared results using C4 as the calibration set against those using I.I.D downstream task datasets as the calibration set. We found that I.I.D downstream tasks as calibration sets do not outperform high-quality pretraining corpora as calibration sets, with both yielding comparable performance. This further supports our hypothesis that, unlike fields such as CV, highly relevant data as calibration sets do not significantly enhance performance.

---

> ### Author Response · Authors · 2024-11-25
>
> > Q2: For W2, in the past research, it has been proven that larger LLMs are less sensitive to quantization. Therefore, due to the time constraint of the rebuttal, there is no need to extend the experimental models to various sizes. It suffices to provide experimental data on the largest model (e.g., llama2-70B) to demonstrate that the current conclusions remain valid.
>
> Thank you for your valuable suggestion!
>
> The point you raised about 70B+ models is highly relevant. In this paper, we primarily focused on models in the 7B-13B range to highlight key concepts that we believe are broadly applicable to models of various sizes. However, we acknowledge that as model sizes scale up, new patterns and behaviors may arise. We intend to investigate the performance of larger models in our future research and plan to incorporate experiments with the 70B model in the final version of the paper.

---

> ### Author Response · Authors · 2024-11-25
>
> As a kind reminder, the discussion period is drawing close. Please let us know if there remains anything that we can further clarify to improve our work. Many thanks in advance.

---

> > ### Comment · Reviewer_a5T8 · 2024-12-02
> >
> > Thank you for the rebuttal. However, I do not believe my concerns about the lack of in-depth analysis and absence of experiments on larger models have been adequately addressed. Therefore, I will maintain my score.

---

> > > ### Author Response · Authors · 2024-12-03
> > >
> > > For the comparison of results between the I.I.D. and OOD settings, we conducted a more in-depth analysis. We performed the Wilcoxon signed-rank test on the data in Table 2 and examined whether there were significant differences between the paired samples of the I.I.D. and OOD datasets.
> > >
> > > Null hypothesis (H₀): There is no significant difference in performance between the I.I.D and OOD settings.
> > >
> > > Alternative hypothesis (H₁): There is a significant difference in performance between the I.I.D and OOD settings.
> > >
> > > By setting the significance level ( α = 0.05), we calculated the p-values for each method and each task, as shown in the table below:
> > >
> > > | Algorithm\Dataset |  EQA  |  SA   |  NLI  |  TD   |
> > > |:-----------------:|:-----:|:-----:|:-----:|:-----:|
> > > |       GPTQ        | 0.433 | 0.252 | 0.155 | 0.821 |
> > > |       SpQR        | 0.860 | 0.940 | 0.016 | 0.348 |
> > > |        AWQ        | 0.850 | 0.850 | 0.594 | 0.669 |
> > > |    Smoothquant    | 0.075 | 0.612 | 0.893 | 0.286 |
> > >
> > > We can observe that all the p-values are nearly greater than the significance level α, indicating that we fail to reject the null hypothesis (H₀), and **there is no significant difference in performance between the I.I.D and OOD settings**.
> > >
> > > In addition, we have conducted an in-depth analysis in **Section 3.3**, which includes summarizing conclusions, proposing hypotheses, and performing validations.
> > >
> > > Regarding your suggestion to experiment with larger models, it is important to note that current quantization evaluation work is not entirely focused on large models[1,2]. Our research primarily emphasizes distribution shift experiments. While we recognize that more comprehensive evaluations are valuable for benchmarking, we are currently limited by computational resources and are unable to conduct experiments on larger models at this stage. However, we plan to continue expanding our experiments to include larger models when feasible.
> > >
> > > [1] Work in Progress COMPRESSING LLMS: THE TRUTH IS RARELY PURE AND NEVER SIMPLE
> > >
> > > [2] How Does Calibration Data Affect the Post-training Pruning and Quantization of Large Language Models?

---

### Official Review · Reviewer_jPYg · 2024-11-03

**Soundness:** 2
**Presentation:** 3
**Contribution:** 2
**Rating:** 5
**Confidence:** 2

**Summary:**

The authors proposed a benchmark for evaluating the post-training quantized large language models (LLMs) generalization ability. They considered two scenarios and utilized 40 datasets. Additionally, they released a modular-designed toolbox.

**Strengths:**

The authors conducted comprehensive experiments, providing meaningful results that highlight the impact of calibration data on post-training quantization accuracy.

**Weaknesses:**

compared to post-training quantization, the influence of data on quantization finetuning methods, such as Q-LoRA, is more significant. This is because the calibration data for post-training quantization is limited, making the model more susceptible to overfitting and data influence in Q-LoRA.

Regarding the accuracy numbers presented in the table, it's important to know whether they represent a single trial or are averaged across multiple trials. Quantized networks can exhibit variance, and relying on a single trial may not provide reliable guidance. It is crucial to comprehend the inherent variability of a specific PTQ method prior to drawing conclusions regarding the impact of data on quantization accuracy.

**Questions:**

Did the authors experiment with different samples from the C4 dataset?
Did authors measure variance even when using the same dataset, like C4, but with different examples?
Understanding these aspects would provide deeper insights into the robustness and reliability of the quantization process.

In line 1249, the authors mentioned: We present the average results with random seeds 42 and 567. Why particular choose 42 and 567 as random seed? What if we use other random seeds, like 0 or 1?

---

> ### Author Response · Authors · 2024-11-13
>
> > Q: Did the authors experiment with different samples from the C4 dataset? Did authors measure variance even when using the same dataset, like C4, but with different examples? Understanding these aspects would provide deeper insights into the robustness and reliability of the quantization process.
> In line 1249, the authors mentioned: We present the average results with random seeds 42 and 567. Why particular choose 42 and 567 as random seed? What if we use other random seeds, like 0 or 1?
>
> Thank you for your question and suggestion!
>
> In our experiments, we used C4 as the calibration set for the S1 scenario. For this experiment, we selected two random seeds, 42 and 567, to perform two different samplings on C4, with the results shown in **Figure 6**. By comparing the outcomes from the two different random seeds, we found that the experimental results showed almost no variation, aside from a few isolated cases.
>
> To avoid further increasing our already extensive workload, we initially conducted the experiment with only two random seeds. However, to ensure rigor, we are currently expanding the experiment with additional random seeds, such as 0 or 1.

---

> > ### Author Response · Authors · 2024-11-23
> >
> > The following are the results obtained with a random seed of 0, using the GPTQ method for experimentation on the SA task in BOSS.
> >
> > | 0-shot | Bit |    AZ     |    DS     |    SE     |  SST  |
> > |:------:|:---:|:---------:|:---------:|:---------:|:-----:|
> > |   AZ   |  4  |   58.37   | **84.45** |   78.12   | 55.48 |
> > |   AZ   |  3  | **10.56** |   0.08    |   0.07    | 5.67  |
> > |   DS   |  4  |   35.20   | **51.14** |   37.76   | 35.0  |
> > |   DS   |  3  | **5.49**  |   0.00    |   0.07    | 1.62  |
> > |   SE   |  4  |   1.78    | **32.42** |   24.15   | 16.66 |
> > |   SE   |  3  |   1.01    |   0.00    | **35.48** | 1.18  |
> > |  SST   |  4  |   47.59   | **56.58** |   45.37   | 45.37 |
> > |  SST   |  3  |   0.52    |    0.0    | **43.81** | 1.04  |
> >
> > | 3-shot | Bit |    AZ     |    DS     |  SE   |  SST  |
> > |:------:|:---:|:---------:|:---------:|:-----:|:-----:|
> > |   AZ   |  4  |   67.36   | **84.65** | 62.19 | 78.21 |
> > |   AZ   |  3  | **87.44** |   71.32   | 59.20 | 41.39 |
> > |   DS   |  4  |   48.53   | **59.28** | 44.93 | 54.55 |
> > |   DS   |  3  | **57.64** |   36.64   | 38.26 | 51.07 |
> > |   SE   |  4  |   40.92   | **45.16** | 42.32 | 41.95 |
> > |   SE   |  3  | **47.33** |   38.28   | 43.57 | 42.21 |
> > |  SST   |  4  |   44.07   | **61.54** | 31.81 | 54.50 |
> > |  SST   |  3  | **54.89** |   51.76   | 32.33 | 36.90 |
> >
> > We can still observe that, for the same test dataset, it is not necessarily the case that using an I.I.D. dataset as the calibration set yields superior performance. Due to time constraints, we have only presented partial results here. Moving forward, we will test additional samplings to ensure the generalizability and robustness of the experiment.

---

> ### Author Response · Authors · 2024-11-23
>
> > W：Compared to post-training quantization, the influence of data on quantization finetuning methods, such as Q-LoRA, is more significant. This is because the calibration data for post-training quantization is limited, making the model more susceptible to overfitting and data influence in Q-LoRA.
>
> Thank you for your comments!
>
> Efficient inference methods like Q-LoRA, which involve quantizing the model before fine-tuning, entail two sets of data: one for quantization and one for fine-tuning. Along with the data from the pre-training and testing stages, this results in a total of four data components. This makes the distribution shift process highly complex, making it difficult to isolate and decouple the impact of calibration-stage data on final performance. Therefore, this aspect is beyond the scope of our current discussion but could be considered as a direction for future work.
>
> In addition, some research [1] suggests that providing prompts to LLMs effectively performs implicit fine-tuning on the model. Similarly, the PTQ method we are testing, when given an LLM prompt, also implicitly fine-tunes the model, rather than explicitly doing so.
>
> [1] Why can GPT learn in-context? language models secretly perform gradient descent as meta-optimizers

---

> ### Author Response · Authors · 2024-11-25
>
> As a kind reminder, the discussion period is drawing close. Please let us know if there remains anything that we can further clarify to improve our work. Many thanks in advance.

---

### Meta-Review · Area_Chair_71WC · 2024-12-20

**Metareview:**

This paper evaluates the generalization ability of quantized LLMs through a benchmark suite, offering an evaluation system, detailed analyses, and a modular toolbox. The study examines the impact of calibration data distribution on quantized LLMs using over 40 datasets and popular LLMs like LLaMA and Baichuan.

Concerns were raised about the lack of in-depth analysis and the superficial treatment of experimental findings. The paper does not provide clear guidance on selecting calibration data to enhance generalization, limiting its practical impact. Visualization issues, such as unclear radar charts, were also noted.

The reviewers had mixed opinions, with some recognizing the potential of the work, while others were skeptical due to the lack of depth and practical implications. The paper shows promise but needs refinement, particularly in analysis depth and practical data selection guidance.

This paper received an averaged score of 4.75, which, while above the threshold, is not competitive among this year's submissions. Given the balance of strengths and weaknesses, the final recommendation is to reject this submission in its current form.

**Additional Comments On Reviewer Discussion:**

In the rebuttal period, the central discussions among reviewers focused on the superficial nature of the analysis and the lack of practical guidance provided by the paper regarding the selection of calibration data for quantized LLMs. Reviewers expressed concerns that the findings, while novel, did not offer substantial advancements in the field due to these shortcomings. The authors made efforts to address these issues by supplying additional statistical tests and attempting to clarify their findings. However, the overall depth and practical applicability of the paper remained insufficient.

In my final decision, I took into account the authors' engagement with the reviewers' comments and their attempts to bolster the paper's contributions. Despite the authors' efforts to respond to the concerns raised, the paper still fell short in providing the necessary depth of analysis and concrete recommendations for practical application. As a result, I concluded that the paper, in its current form, does not meet the standards for acceptance. The decision to reject was based on the persistent gaps in the analysis and the lack of actionable insights, which are critical for the paper to have a significant impact on the field of quantized LLMs.

---

### Decision · Program_Chairs · 2025-01-22

Reject